# OLion: Approaching the Hadamard Ideal by Intersecting Spectral and $\ell_\infty$ Implicit Biases

Zixiao Wang [*1]   Yifei Shen [*2]   Huishuai Zhang [*1]

## Abstract

Many optimizers can be interpreted as steepest-descent methods under norm-induced geometries, and thus inherit corresponding implicit biases. We introduce *OLion* (*Orthogonal Lion*), which combines spectral control from orthogonalized update directions with $\ell_\infty$-style coordinate control from sign updates. *OLion* forms a Lion-style momentum direction, approximately orthogonalizes it via a few Newton–Schulz iterations, and then applies an entrywise sign, providing an efficient approximation to taking a maximal step over the intersection of the spectral and $\ell_\infty$ constraint sets (a scaled Hadamard-like set for matrix parameters). Despite the strong nonlinearity of orthogonalization and sign, we prove convergence under a mild, empirically verified diagonal-isotropy assumption. Across large-scale language and vision training, including GPT-2 and Llama pretraining, SiT image pretraining, and supervised fine-tuning, *OLion* matches or outperforms AdamW and Muon under comparable tuning while using only momentum-level optimizer state, and it mitigates optimizer mismatch when fine-tuning AdamW-pretrained checkpoints.

**Code:** https://github.com/kv-wang/OLion

## 1. Introduction

In addition to wall-clock efficiency and stability, the choice of optimizer shapes *which* solutions are found through its *implicit bias* (Gunasekar et al., 2018; Soudry et al., 2018; Li et al., 2023b; Wang et al., 2021a; Bernstein et al., 2018; Vasudeva et al., 2024; Lyu & Li, 2020). A useful unifying view is that many first-order methods implement (approximately)

---
[*]Equal contribution   [1]Peking University, Beijing, China. [2]Microsoft Research Asia, Beijing, China. Correspondence to: Huishuai Zhang <zhanghuishuai@pku.edu.cn>.

*Proceedings of the 43$^{rd}$ International Conference on Machine Learning*, Seoul, South Korea. PMLR 306, 2026. Copyright 2026 by the author(s).

*Figure 1.* The geometry motivation: We view Muon and Lion as maximal-update methods under two norm-induced geometries: a spectral geometry (orthogonalization / polar factor) and an $\ell_\infty$ geometry (sign-based coordinate normalization). Their intersection suggests a scaled Hadamard set as an idealized target for matrix-shaped updates, motivating an intersection-seeking design.

steepest descent under different norm-induced geometries, yielding different regularization effects even when the explicit objective is identical (Bernstein & Newhouse, 2024).

Figure 1 illustrates the geometric motivation that guides this work. Two recent optimizers highlight complementary biases. *Muon* uses a structured update for matrix-shaped parameters: it orthogonalizes the momentum direction using a small number of Newton–Schulz iterations, producing a direction-only update with a flattened singular-value profile. From the implicit-bias viewpoint, Muon promotes a form of spectral control by limiting amplification along singular directions. In contrast, *Lion* emphasizes a different geometry: it applies an element-wise sign to a momentum direction. The sign operation caps each coordinate's contribution and behaves like steepest descent under an $\ell_\infty$ constraint.

These developments motivate a natural question. Muon provides strong *global* control through spectral normalization but lacks element-wise normalization; Lion provides strong *coordinate-wise* control but does not enforce spectral structure. *Can we obtain the benefits of both spectral and $\ell_\infty$*

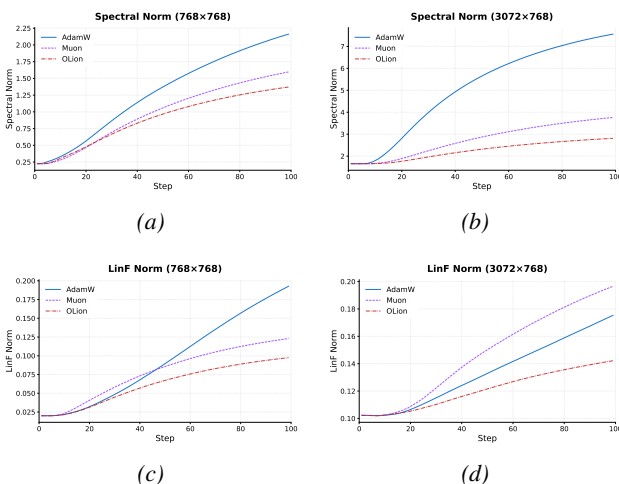

*Figure 2.* Evolution of spectral and $\ell_\infty$ norms during GPT-2 small pretraining for weight matrices of different shapes. Adam favors a small $\ell_\infty$ norm, Muon favors a small spectral norm, while *OLion* simultaneously exhibits both implicit biases.

*implicit biases in a single optimizer?* A related practical motivation is the *pretrain–fine-tuning mismatch*: many widely used pretrained checkpoints were produced with optimizers that include element-wise normalization (e.g., AdamW-like behavior), and optimizer mismatch can measurably affect fine-tuning dynamics. For example, recent studies show that Muon fine-tuning of models pretrained with Adam performs worse than its Adam counterpart (Liu et al., 2025a).

We approach this question through the geometry of intersecting constraints (Figure 1). For matrix parameters, Muon's orthogonalization encourages updates that lie near a scaled orthogonal set (a spectral-norm geometry), while Lion's sign update encourages proximity to the $\ell_\infty$ extreme set (entries with equal magnitude). For square matrices, the intersection of these two structures corresponds to a scaled *Hadamard set*, i.e., matrices with orthogonal rows/columns and entries $\pm 1/\sqrt{d}$. Although the Hadamard ideal may not be literally solvable, the perspective is useful: composing an orthogonalization map with a sign map resembles an approximate intersection-seeking procedure.

Motivated by the view, we propose *OLion* (*Orthogonal Lion*), which integrates Muon's Newton–Schulz orthogonalization with a Lion-style sign update. At each step, *OLion* forms a momentum direction, orthogonalizes this direction, and then applies an element-wise sign to obtain a coordinate-normalized update. In the implementation, we optionally apply a lightweight magnitude alignment (e.g., Root Mean Squares (RMS) scaling) stabilizes effective step sizes across layers and tensor shapes. As a result, *OLion* preserves Muon's memory efficiency while incorporating the practical benefits of sign-based updates.

Beyond optimization, the sign operation in *OLion* brings two

practical systems-level benefits. First, sign updates naturally support communication-efficient multi-node training: when the update direction is reduced to its element-wise sign, the communicated message can be heavily compressed (e.g., to 1-bit per entry, optionally with a shared scale), which can reduce bandwidth pressure in distributed data-parallel settings (Liu et al., 2024a). Second, the $\ell_\infty$-style implicit bias induced by sign updates promotes more uniform coordinate magnitudes, leading to weight and update tensors with a flatter entrywise distribution. Such uniformity is desirable for low-precision deployment, as quantization error is often dominated by a few large-magnitude outliers; suppressing these outliers can improve effective quantization fidelity at a fixed bit-width (Chee et al., 2023; Tseng et al., 2024; Liu et al., 2024b; Ashkboos et al., 2024).

A simple experiment indeed confirms the intended bias intersection: *OLion* maintains both a small spectral norm and a small $\ell_\infty$ norm during training, whereas other optimizers favor only one of the two, as shown in Figure 2.

**Contributions.** Our contributions span formulation, algorithm design, theory, and large-scale evaluation. First, we provide a norm-geometric formulation that explicitly intersects spectral control (via orthogonalization) with $\ell_\infty$-style coordinate control (via sign), yielding a Hadamard-type ideal for matrix-shaped updates. Second, guided by this formulation, we design *OLion* as an effective and efficient optimizer: it applies sign-after-orthogonalization to a Lion-style momentum direction, with stable default scaling rules. Third, we establish convergence guarantees under a new assumption, supported by theoretical justification and empirical verification. Finally, we validate *OLion* at scale across text and vision and across training stages, including GPT-2 and Llama pretraining, SiT image pretraining, and Llama supervised fine-tuning, showing consistent improvements over strong baselines.

## 2. Related Work

Implicit bias, i.e., how the optimizer's geometry selects solutions among many minimizers, has been an essential lens for understanding generalization in over-parameterized models (Soudry et al., 2018; Gunasekar et al., 2018; Lyu & Li, 2020; Vasudeva et al., 2024; Zhang et al., 2024a; Wang et al., 2021a;b). Our work leverages this spirit by combining two complementary geometries *within a single update*: spectral control via orthogonalization and $\ell_\infty$-style coordinate control via sign. While prior work has studied each geometry in isolation (and sometimes contrasted them), *OLion* is motivated by their intersection for matrix-shaped parameters, which corresponds to a Hadamard-type ideal.

## 2.1. Spectral geometry and orthogonalized updates

**Orthogonality-constrained optimization.** Optimization over the Stiefel/orthogonal manifolds is a classical topic in matrix manifold optimization. Many first-order methods proceed by taking an ambient descent step and then projecting (or retracting) back to the feasible set (Absil et al., 2008; Edelman et al., 1998; Wen & Yin, 2013; Gao et al., 2018; Li et al., 2020). A standard retraction uses the polar factor: for full-rank $\boldsymbol{A}$, the orthogonal factor $\mathcal{O}(\boldsymbol{A}) = \boldsymbol{A}(\boldsymbol{A}^\top \boldsymbol{A})^{-1/2}$ (equivalently $\boldsymbol{U}\boldsymbol{V}^\top$ from the SVD) is the closest orthogonal matrix to $\boldsymbol{A}$ in Frobenius norm (Higham, 1986). This polar-based retraction is widely used in practical algorithms and often enables clean descent arguments under smoothness assumptions (Absil et al., 2008; Wen & Yin, 2013; Gao et al., 2018).

**Optimizers as steepest descent under norm-induced geometries.** Recent frameworks reinterpret modern optimizers as (approximate) steepest descent under different norm constraints (Bernstein & Newhouse, 2024). Within this view, Shampoo (Gupta et al., 2018) can be seen as operating in a spectral geometry via preconditioning, and Muon (Jordan et al., 2024b) explicitly orthogonalizes momentum updates using Newton–Schulz iterations, yielding direction-only updates. Muon has been shown to be effective and memory-efficient for large-scale training (Liu et al., 2025a).

**Muon variants.** Several follow-up works extend Muon in different directions. Muon-MVR (Chang et al., 2025) improves stochastic convergence by adding momentum-based variance reduction. MuonMax (Goldstein et al., 2025) targets robustness to learning-rate selection by aggregating norms across layers. Turbo-Muon (Boissin et al., 2025) reduces the overhead of orthogonalization, while 8-bit Muon (Gupta et al., 2025) targets optimizer-state memory. AdaMuon (Si et al., 2025) augments orthogonal updates with element-wise second-moment statistics; NorMuon (Li et al., 2025) introduces neuron-level adaptive learning rates alongside orthogonalization. MuonAll (Page et al., 2025) removes the hybrid-training dependency by reshaping 1D parameters (e.g., biases, normalization parameters) into 2D forms so that a spectral update rule can be applied uniformly.

## 2.2. $\ell_\infty$ geometry and sign-based updates

**Coordinate-wise adaptivity and Adam-style methods.** Adam (Kingma & Ba, 2014; Loshchilov & Hutter, 2019) and related diagonal preconditioned methods (e.g., Ada-Grad (Duchi et al., 2011), RMSProp) normalize gradients coordinate-wise through an adaptive diagonal scaling. In norm-geometric terms, diagonal preconditioning can be interpreted as steepest descent under a coordinate-weighted $\ell_\infty$ geometry (Bernstein & Newhouse, 2024; Xie & Li, 2024). A practical limitation is optimizer-state memory due to second-moment statistics, motivating memory-efficient

variants that approximate or compress these estimates, including Adafactor (Shazeer & Stern, 2018), SM3 (Anil et al., 2019), CAME (Luo et al., 2023), AdaLomo (Lv et al., 2023), and blockwise schemes such as Adam-mini (Zhang et al., 2024b), as well as recent progress that remove explicit second-moment state such as AdamS (Zhang et al., 2025).

**Sign and quantized updates.** Sign-based updates are also a classical extreme in gradient compression and quantization. SIGNSGD (Bernstein et al., 2018) replaces gradients by their entry-wise signs, yielding strong communication savings together with convergence guarantees under standard stochastic assumptions. The majority-vote variant improves robustness in multi-worker settings (Bernstein et al., 2019). More broadly, quantized-gradient methods and error-feedback mechanisms provide principled ways to compress updates while retaining convergence (Alistarh et al., 2017; Karimireddy et al., 2019; Bernstein et al., 2018; 2019).

**Lion and $\ell_\infty$ implicit bias.** The Lion (Evolved Sign Momentum) optimizer (Chen et al., 2023) combines a lightweight momentum mechanism with a sign update and removes Adam's second-moment accumulator, reducing optimizer-state memory and often improving throughput. Follow-up theory connects Lion to constrained optimization and $\ell_\infty$ geometry interpretations (Chen et al., 2025) and (Sfyraki et al., 2025) unifies Lion and Muon under a stochastic Frank–Wolfe viewpoint, where Lion corresponds to an $\ell_\infty$ geometry while Muon operates on a spectral ball. Convergence analyses for Lion in non-convex stochastic settings and distributed regimes have also been developed (Jiang & Zhang, 2025), and smooth variants such as RLion replace $\text{sign}(\cdot)$ with a smooth transformation to mitigate potential instability (Rong et al., 2025).

**Positioning of OLion.** OLion is motivated by the intersection of two geometries, and the corresponding feasible update sets for matrix-shaped parameters lead to a Hadamard-type ideal. OLion approaches such intersection in a single update while retaining a momentum-level memory footprint.

## 3. Preliminaries

**Notation.** We consider a matrix parameter $\boldsymbol{X}_t \in \mathbb{R}^{d_1 \times d_2}$ with $d_1 \geq d_2$. Let $\boldsymbol{G}_t := \nabla f(\boldsymbol{X}_t)$ denote the (full) gradient at step $t$, and let $\widetilde{\boldsymbol{G}}_t$ denote the update signal actually used by the optimizer (e.g., a momentum/Nesterov combination of mini-batch gradients). We use the Frobenius inner product $\langle \boldsymbol{A}, \boldsymbol{B} \rangle := \text{tr}(\boldsymbol{A}^\top \boldsymbol{B})$. For models with multiple matrix-valued parameters, we write $\overrightarrow{\boldsymbol{X}} := \{\boldsymbol{X}_{(1)}, \boldsymbol{X}_{(2)}, \ldots, \boldsymbol{X}_{(L)}\}$. Since the update rule is applied separately to each parameter block and the analysis is also separable across matrices, we present the algorithm and theory for a single matrix for clarity; extending the statements

to $\overrightarrow{\boldsymbol{X}}$ is straightforward but requires heavier notation.

**Update geometry and implicit bias.** Many first-order optimizers can be viewed as choosing an update direction according to a particular update geometry, which in turn induces an implicit bias on the solutions reached. In this work, we focus on two geometries that are especially natural for matrix-shaped parameters: spectral control (orthogonalized directions) and $\ell_\infty$-style coordinate control (sign directions).

**Two structured sets and the Hadamard ideal (tall case).** To combine these two biases for $\boldsymbol{X} \in \mathbb{R}^{d_1 \times d_2}$ with $d_1 \geq d_2$, we define

$$\mathcal{A} := \left\{ \boldsymbol{X} \in \mathbb{R}^{d_1 \times d_2} : \boldsymbol{X}^\top \boldsymbol{X} = \boldsymbol{I}_{d_2} \right\}, \qquad (1)$$

$$\mathcal{B} := \left\{ \boldsymbol{X} \in \mathbb{R}^{d_1 \times d_2} : \boldsymbol{X}_{ij} \in \left\{ \pm \frac{1}{\sqrt{d_1}} \right\} \right\}. \qquad (2)$$

The set $\mathcal{A}$ captures the spectral extreme (column-orthonormal updates), while $\mathcal{B}$ captures the $\ell_\infty$ extreme (uniform-magnitude, sign-pattern updates). Their intersection $\mathcal{A} \cap \mathcal{B}$ is the set of (scaled) partial Hadamard matrices (when they exist), i.e., matrices with orthonormal columns and entries $\pm 1/\sqrt{d_1}$. We refer to this intersection as the *Hadamard ideal*.

# 4. Method: Orthogonal Lion

Let $\widetilde{\boldsymbol{G}}_t \in \mathbb{R}^{d_1 \times d_2}$ denote the matrix-shaped update signal at step $t$ (e.g., a momentum-smoothed gradient; see Algorithm 1). Guided by the Hadamard ideal introduced in Section 3, we seek an update direction that is simultaneously close to the spectral extreme $\mathcal{A}$ and the $\ell_\infty$ extreme $\mathcal{B}$. A natural idealized target is the maximizer of the linear score over the intersection:

$$\boldsymbol{X}_t^\star \in \arg \max_{\boldsymbol{X} \in \mathcal{A} \cap \mathcal{B}} \left\langle \boldsymbol{X}, \widetilde{\boldsymbol{G}}_t \right\rangle. \qquad (3)$$

Since $\mathcal{A} \cap \mathcal{B}$ is highly nonconvex, solving Equation (3) exactly is intractable. Instead, we approximate it by using alternating-projection steps onto $\mathcal{A}$ and $\mathcal{B}$, which yields a simple and efficient update rule.

**Projection onto $\mathcal{A}$ (Orthogonalization).** The maximizer of $\langle \boldsymbol{X}, \boldsymbol{Z} \rangle$ over $\boldsymbol{X} \in \mathcal{A}$ is given by:

$$P_{\mathcal{A}}(\boldsymbol{Z}) \in \arg \max_{\boldsymbol{X} \in \mathcal{A}} \langle \boldsymbol{X}, \boldsymbol{Z} \rangle \implies P_{\mathcal{A}}(\boldsymbol{Z}) = \mathcal{O}(\boldsymbol{Z}), \quad (4)$$

where if $\boldsymbol{Z} = \boldsymbol{U}\boldsymbol{\Sigma}\boldsymbol{V}^\top$ is the (thin) SVD, then $\mathcal{O}(\boldsymbol{Z}) = \boldsymbol{U}\boldsymbol{V}^\top$ and $\mathcal{O}(\boldsymbol{Z})^\top \mathcal{O}(\boldsymbol{Z}) = \boldsymbol{I}_{d_2}$.

**Projection onto $\mathcal{B}$ (entrywise sign).** Likewise, maximizing $\langle \boldsymbol{X}, \boldsymbol{Z} \rangle$ over $\boldsymbol{X} \in \mathcal{B}$ decouples entrywise and yields

$$P_{\mathcal{B}}(\boldsymbol{Z}) \in \arg \max_{\boldsymbol{X} \in \mathcal{B}} \langle \boldsymbol{X}, \boldsymbol{Z} \rangle \implies P_{\mathcal{B}}(\boldsymbol{Z}) = \frac{\text{sign}(\boldsymbol{Z})}{\sqrt{d_1}}, \quad (5)$$

where $\text{sign}(\cdot)$ is applied entrywise.

## 4.1. The OLion Algorithm

---
**Algorithm 1** *OLion* Optimizer

---
**Require:** Learning rate $\eta_t$, $\beta_1$, $\beta_2$, weight decay $\lambda$, Newton–Schulz steps $K$
**Require:** Initial parameters $\boldsymbol{X}_0$, momentum $\boldsymbol{M}_0 = \boldsymbol{0}$
   **for** $t = 0, 1, \ldots, T-1$ **do**
      Compute mini-batch gradient $\mathbf{g}_t$ (for $\mathbf{G}_t = \nabla f(\mathbf{X}_t)$)
      **Momentum:** $\mathbf{M}_t = \beta_2 \mathbf{M}_{t-1} + (1-\beta_2)\mathbf{g}_t$
      **Nesterov mix:** $\widetilde{\mathbf{G}}_t = (1-\beta_1)\mathbf{g}_t + \beta_1 \mathbf{M}_t$
      **Orthogonalize:** $\mathbf{Q}_t = \text{NEWTONSCHULZ}(\widetilde{\mathbf{G}}_t, K)$
      **Sign operation:** $\mathbf{S}_t = \text{sign}(\mathbf{Q}_t)$
      **RMS alignment:** $\mathbf{D}_t = \gamma_t \mathbf{S}_t$
      **Update:** $\mathbf{X}_{t+1} = \mathbf{X}_t - \eta_t \mathbf{D}_t - \lambda \eta_t \mathbf{X}_t$
   **end for**

---

*OLion* **direction (sign-after-orthogonalization).** Composing two projections Equation (4) and Equation (5) gives the *OLion* direction:

$$P_{\mathcal{B}}\left( P_{\mathcal{A}}(\widetilde{\boldsymbol{G}}_t) \right) = \frac{1}{\sqrt{d_1}} \text{sign}\left( \mathcal{O}(\widetilde{\boldsymbol{G}}_t) \right), \qquad (6)$$

which can be viewed as a one-step approximation to solve Equation (3). Intuitively, $\mathcal{O}(\widetilde{\boldsymbol{G}}_t)$ enforces spectral structure by normalizing singular values, and the subsequent sign enforces coordinate-wise normalization, producing an update direction that simultaneously reflects both implicit biases.

For convenience and consistency with Algorithm 1, we introduce $\boldsymbol{S}_t := \text{sign}(\mathcal{O}(\widetilde{\boldsymbol{G}}_t))$. We next introduce other components in Algorithm 1.

**Multi-timescale momentum.** Algorithm 1 first constructs the update signal as follows, which is the same as Lion,

$$\mathbf{M}_t = \beta_2 \mathbf{M}_{t-1} + (1-\beta_2)\mathbf{g}_t, \qquad \widetilde{\mathbf{G}}_t = (1-\beta_1)\mathbf{g}_t + \beta_1 \mathbf{M}_t.$$

The buffer $\mathbf{M}_t$ aggregates longer-horizon history controlled by the "slow" coefficient $\beta_2$, while the mixing coefficient $\beta_1$ ensures the current mini-batch gradient $\mathbf{g}_t$ contributes a fixed fraction $(1 - \beta_1)$, preserving responsiveness.

**Root Mean Square (RMS) alignment.** Since $\mathbf{S}_t$ has fixed-magnitude entries, implementations often apply a scalar alignment to stabilize effective step sizes across layers and shapes. In Algorithm 1, one option is RMS alignment: $\mathbf{D}_t := \gamma_t \mathbf{S}_t$. Specifically, to maintain compatibility with Adam's learning rate schedules, we scale the RMS norm of the final update to match Adam's empirical RMS value of $\approx 0.2$ (Liu et al., 2025b), yielding

$$\gamma_t = \frac{0.2}{\text{RMS}(\mathbf{S}_t)} = \frac{0.2\sqrt{d_1 d_2}}{\|\mathbf{S}_t\|_F} \approx 0.2, \qquad (7)$$

where $d_1$ and $d_2$ denote the number of rows and columns of the matrix $\mathbf{S}_t$, respectively.

## 4.2. Convergence Analysis of OLion

To establish the convergence analysis of OLion, we first introduce some standard assumptions.

**Assumption 4.1** (Smoothness and lower boundedness)**.** The function $f : \mathbb{R}^{d_1 \times d_2} \to \mathbb{R}$ is differentiable and $L$–smooth with respect to $\|\cdot\|_F$, i.e., for all $\mathbf{X}, \mathbf{Y}$,

$$f(\mathbf{Y}) \le f(\mathbf{X}) + \langle \nabla f(\mathbf{X}), \mathbf{Y} - \mathbf{X} \rangle + \frac{L}{2} \|\mathbf{Y} - \mathbf{X}\|_F^2. \quad (8)$$

Moreover, $f$ is bounded below: $f(\mathbf{X}) \ge f_{inf}$ for all $\mathbf{X}$.

Because the sign-after-orthogonalization operation is highly nonlinear, a direct descent analysis for OLion is generally impossible without leveraging additional structure in the gradient. To make the analysis tractable, we impose the following assumption on the singular-vector geometry of the update signal. Throughout the convergence proof, we assume that the $\widetilde{\mathbf{G}}_t$ used in Algorithm 1 satisfies this condition at every iteration.

**Assumption 4.2** (Diagonal-isotropy decomposition)**.** For a rank-$r$ matrix $\mathbf{Z}$ with singular value decomposition form $\mathbf{Z} = \mathbf{U}\boldsymbol{\Sigma}\mathbf{V}^\top$, with $\mathbf{U} \in \mathbb{R}^{d_1 \times r}$, $\mathbf{V} \in \mathbb{R}^{d_2 \times r}$, $\boldsymbol{\Sigma} = \mathrm{diag}(\sigma_1, \dots, \sigma_r) \succeq 0$. There exists some $\varepsilon \ge 0$ such that,

$$\left\| \mathrm{diag}\left(\mathbf{U}^T \mathrm{sign}(\mathbf{U}\mathbf{V}^\top)\mathbf{V}\right) - \frac{\|\mathbf{U}\mathbf{V}^\top\|_1}{r}\mathbf{1} \right\|_2 \le \varepsilon \frac{\|\mathbf{U}\mathbf{V}^\top\|_1}{\sqrt{r}}.$$

**Intuition for diagonal isotropy.** Let $\mathbf{S} = \mathrm{sign}(\mathbf{U}\mathbf{V}^\top)$. Define

$$m_k := (\mathbf{U}^\top \mathbf{S}\mathbf{V})_{kk} = \mathbf{u}_k^\top \mathbf{S}\mathbf{v}_k = \langle \mathbf{u}_k \mathbf{v}_k^\top, \mathbf{S} \rangle, \quad k \in [r],$$

which measures how strongly the sign pattern $\mathbf{S}$ correlates with the $k$-th rank-one singular-direction component $\mathbf{u}_k \mathbf{v}_k^\top$. Note that

$$\sum_{k=1}^{r} m_k = \mathrm{tr}(\mathbf{U}^\top \mathbf{S}\mathbf{V}) = \langle \mathbf{U}\mathbf{V}^\top, \mathbf{S} \rangle = \|\mathbf{U}\mathbf{V}^\top\|_1,$$

so the average correlation is $\bar{m} = \|\mathbf{U}\mathbf{V}^\top\|_1 / r$.

Therefore, Assumption 4.2 indicates that the $m_k$'s are nearly constant across $k$ (i.e., $\mathrm{diag}(\mathbf{U}^\top \mathbf{S}\mathbf{V}) \approx \bar{m}\,\mathbf{1}$). The diagonal-isotropy assumption rules out situations where the sign pattern $\mathbf{S}$ aligns much more strongly with some singular directions than others.

We note that random matrices satisfy Assumption 4.2 for $\varepsilon = o(1)$ with high probability as the matrix dimension gets larger. We formally state the results in Appendix B. We also empirically verify that along the training trajectory,

the $\widetilde{\mathbf{G}}_t$ satisfy Assumption 4.2 with small values of $\varepsilon$ (see Appendix C).

Under Assumption 4.2, we can prove an important lemma as follows.

**Lemma 4.3** (Cancellation-aware upper bound)**.** *For a rank-$r$ matrix $\mathbf{Z}$ with singular value decomposition form $\mathbf{Z} = \mathbf{U}\boldsymbol{\Sigma}\mathbf{V}^\top$ with $\mathbf{U} \in \mathbb{R}^{d_1 \times r}$, $\mathbf{V} \in \mathbb{R}^{d_2 \times r}$, $\boldsymbol{\Sigma} = \mathrm{diag}(\sigma_1, \dots, \sigma_r) \succeq 0$. Suppose that Assumption 4.2 holds. Let $\alpha := \mathrm{tr}(\boldsymbol{\Sigma})/r$. Then*

$$\left| \langle \mathbf{Z} - \alpha\mathbf{U}\mathbf{V}^\top, \mathrm{sign}(\mathbf{U}\mathbf{V}^\top) \rangle \right| \le \varepsilon \frac{\|\mathbf{U}\mathbf{V}^\top\|_1}{\sqrt{r}} \left\| \boldsymbol{\Sigma} - \alpha\mathbf{I} \right\|_F.$$

*Proof.* The proof is deferred to Appendix D. $\square$

We note that a naïve bound to control $\langle \mathbf{Z} - \alpha\mathbf{Q}, \mathbf{S} \rangle$ is given by Cauchy-Schwarz:

$$\left| \langle \mathbf{Z} - \alpha\mathbf{Q}, \mathbf{S} \rangle \right| \le \|\mathbf{Z} - \alpha\mathbf{Q}\|_F \|\mathbf{S}\|_F,$$

but this bound ignores two key structural facts: (i) $\alpha = \mathrm{tr}(\boldsymbol{\Sigma})/r$ makes $\boldsymbol{\Sigma} - \alpha\mathbf{I}$ *trace* 0, so its positive and negative diagonal deviations must cancel, and (ii) $\mathbf{S} = \mathrm{sign}(\mathbf{Q})$ is not an arbitrary matrix, but is tightly coupled to $\mathbf{Q} = \mathcal{O}(\mathbf{Z})$. In fact, Lemma 4.3 exploits these structures and the upper bound can be orders of magnitude tighter than Cauchy–Schwarz bound when $\mathbf{Q}$ is dense (large $\|\mathbf{Q}\|_1$) and the diagonal correlations are nearly uniform.

Conceptually, the estimate shows that the "sign-after-orthogonalization" direction behaves almost as if $\mathbf{Z}$ were replaced by the isotropic surrogate $\alpha\mathbf{Q}$, with an error controlled by the singular-value spread $\|\boldsymbol{\Sigma} - \alpha\mathbf{I}\|_F$ and the (small) imbalance of diagonal correlations. As a result, the descent analysis depends on meaningful geometric quantities (spectral spread and sign-balance), rather than a pessimistic worst-case bound that treats $\mathbf{S}$ as arbitrary.

**A clean core convergence analysis.** Algorithm 1 contains several implementation components (mini-batch gradient, multi-timescale momentum, and optional RMS alignment) that are useful in practice. However, for the purpose of exposing the main geometric idea behind *OLion* while avoiding heavy notation overloading (e.g., distinguishing $\mathbf{G}_t, \mathbf{M}_t, \widetilde{\mathbf{G}}_t$, and their corresponding orthogonalized/sign variants), we first present the convergence analysis in the deterministic full-gradient setting without momentum and stochasticity. Specifically, in the full-gradient setting, we use $\mathbf{G}_t := \nabla f(\mathbf{X}_t)$ and define

$$\mathbf{Q}_t := \mathcal{O}(\mathbf{G}_t), \quad \mathbf{S}_t := \mathrm{sign}(\mathbf{Q}_t), \quad (9)$$
$$\mathbf{X}_{t+1} \leftarrow \mathbf{X}_t - \eta_t \mathbf{S}_t. \quad (10)$$

The scalar alignment factor (e.g. RMS matching) in Algorithm 1 can be absorbed into the effective learning rate and we use a single $\eta_t$ for convenience.

**Auxiliary quantities.** Let $G_t$ have SVD $G_t = U_t \Sigma_t V_t^\top$ with rank $r_t$ and $\Sigma_t = \mathrm{diag}(\sigma_{t,1}, \ldots, \sigma_{t,r_t})$. Define

$$\alpha_t := \frac{\mathrm{tr}(\Sigma_t)}{r_t}, \qquad \rho_t := \frac{\|\Sigma_t - \alpha_t I\|_F}{\alpha_t \sqrt{r_t}}.$$

Define the OLion stationarity measure

$$\Phi_t := \|Q_t\|_1 \, \alpha_t \big(1 - \varepsilon\rho_t\big), \tag{11}$$

which is the quantity we aim to control.

**Theorem 4.4** (Descent and OLion convergence). *Suppose Assumption 4.1 holds. At each step $t$, assume $G_t = \nabla f(X_t)$ satisfies Assumption 4.2 with parameter $\varepsilon$. Consider the deterministic OLion update $X_{t+1} = X_t - \eta_t S_t$ with $S_t = \mathrm{sign}(\mathcal{O}(G_t))$. Then for all $t$,*

$$f(X_{t+1}) \le f(X_t) - \eta_t \Phi_t + \frac{L}{2}\eta_t^2 d_1 d_2, \tag{12}$$

*where $\Phi_t$ is defined in Equation (11). Consequently, for any $T \ge 1$,*

$$\sum_{t=0}^{T-1} \eta_t \Phi_t \le f(X_0) - f_{\inf} + \frac{L}{2} d_1 d_2 \sum_{t=0}^{T-1} \eta_t^2. \tag{13}$$

**Discussion and interpretation.** Theorem 4.4 provides a standard smoothness-based descent guarantee for deterministic OLion, but with a *geometry-aware* stationarity measure

$$\Phi_t = \|Q_t\|_1 \, \alpha_t \big(1 - \varepsilon\rho_t\big), \qquad Q_t = \mathcal{O}(\nabla f(X_t)).$$

Intuitively, the factor $\|Q_t\|_1$ in $\Phi_t$ captures the "density" of the singular spaces of $Q_t$ (Hadamard-like behavior leads to a large $\ell_1$ norm), while $\alpha_t$ measures the average singular value of $G_t$, and the term $(1-\varepsilon\rho_t)$ accounts for the deviation from diagonal isotropy and the spread of singular values.

To connect this OLion-specific measure to the *typical* first-order stationarity criterion $\|\nabla f(X_t)\|_F \to 0$, we add two mild structural conditions: (i) a dense polar factor lower bound on $\|Q_t\|_1$, i.e., $\|Q_t\|_1 \approx \Theta(\sqrt{rd_1d_2})$ and (ii) a spectral-flatness condition that controls the singular-value spread of $\nabla f(X_t)$. Under these conditions, $\Phi_t$ is lower bounded by a positive constant times $\|\nabla f(X_t)\|_F$. Theorem 4.4 yields the finite-time bound

$$\frac{1}{T}\sum_{t=0}^{T-1} \|\nabla f(X_t)\|_F \le \frac{\sqrt{L}\big(f(X_0) - f_{\inf}\big)}{c_\star \sqrt{T}},$$

for some constant $c_\star$, which forces the average gradient norm to vanish at rate $O(1/\sqrt{T})$, matching typical nonconvex bound.

Finally, we note that the analysis for the full gradient can be extended to incorporate (i) momentum/Nesterov mixing (treating $\widetilde{G}_t$ as a biased gradient surrogate controlled

by $(\beta_1, \beta_2)$) and (ii) stochastic gradients (replacing $G_t$ by an unbiased estimator and handling the additional variance terms) using standard techniques from nonconvex optimization (Wang et al., 2023; 2024; Li et al., 2023a). Such extensions are conceptually routine but lead to substantially denser presentations.

# 5. Experiments

We evaluate *OLion* against AdamW and Muon across language and vision, and in both pretraining and supervised fine-tuning settings. Within each setting, we hold the architecture, data, and training pipeline fixed and vary only the optimizer and its hyperparameters. Unless otherwise stated, we use standard defaults for AdamW and Muon and our default $(\beta_1, \beta_2)$ for *OLion*; full configuration details are provided in Appendix A.

## 5.1. *OLion* performs well on LM pretraining

We evaluate *OLion* on standard large-scale language-model pretraining setups. To ensure a fair comparison, we (i) use each baseline optimizer with its *default* hyperparameter configuration as recommended in prior work, and (ii) match the overall update magnitude across methods so that improvements are attributable to the *update geometry* rather than trivially larger steps. Concretely, we first pretrain GPT-2 models (124M–770M) for 48B tokens (100K iterations) following the default AdamW configuration, and then train Llama-2-7B for 32B tokens (8K iterations with batch size 4M) using the default hyperparameters for this regime.

**GPT-2 pretraining (124M–770M).** We study scalability on GPT-2 models trained on OpenWebText at three sizes: *small* (124M), *medium* (355M), and *large* (770M). Figure 3 reports training-loss trajectories. Across all model sizes, *OLion* converges faster than both AdamW, Lion and Muon. This suggests that combining spectral control (via orthogonalization) with coordinate-wise control (via the sign update) improves optimization speed in this canonical pretraining setting.

**Llama-2-7B pretraining.** We next evaluate billion-parameter pretraining with Llama-2-7B by using the FSDP training pipeline. Figure 4 shows both training and validation loss. *OLion* maintains consistently lower loss throughout training compared to Muon, Lion and AdamW, demonstrating that the gains observed on GPT-2 persist at the 7B scale under distributed training.

## 5.2. *OLion* enjoys a wide learning-rate range

We further examine how sensitive *OLion* is to the choice of learning rate. Using the GPT-2 small pretraining setup from Section 5.1, we vary *only* the learning rate and keep

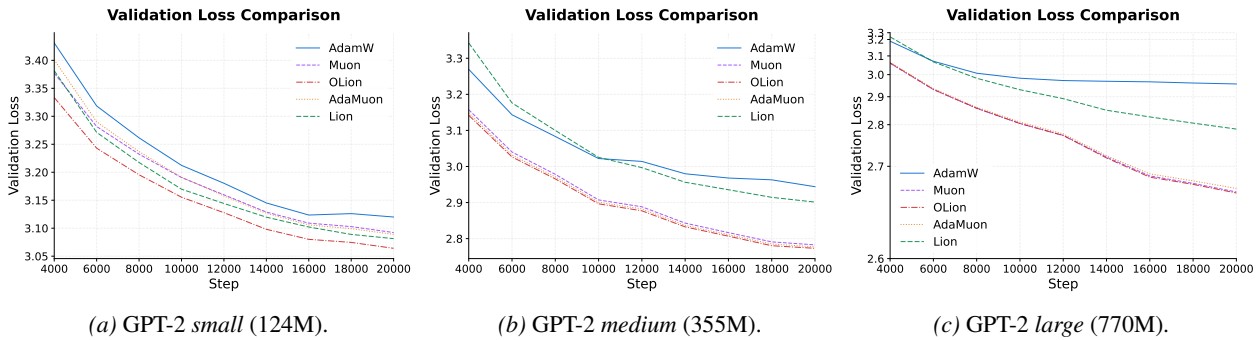

*(a)* GPT-2 *small* (124M).  *(b)* GPT-2 *medium* (355M).  *(c)* GPT-2 *large* (770M).

*Figure 3.* Validation losses for GPT-2 pretraining with AdamW, Lion, Muon, and *OLion*. *OLion* converges faster across all model sizes.

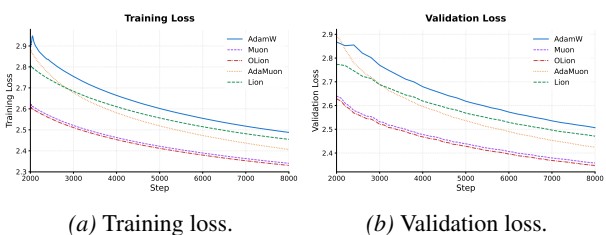

*(a)* Training loss.  *(b)* Validation loss.

*Figure 4.* Llama-2-7B pretraining curves with different optimizers: AdamW, Lion, Muon, and *OLion*.

all other hyperparameters and training procedures fixed. We compare *OLion* against Muon, Lion and AdaMuon under four learning rates: $3 \times 10^{-4}$, $1 \times 10^{-3}$, $2 \times 10^{-3}$, and $5 \times 10^{-3}$. For each optimizer–learning-rate pair, we report the validation loss at training step 10,000.

As shown in Figure 5, *OLion* achieves consistently lower validation loss than other optimizers across the entire range of learning rates. This indicates that the improvements from *OLion* are not tied to a narrowly tuned step size; instead, its advantage persists across substantially different learning-rate regimes.

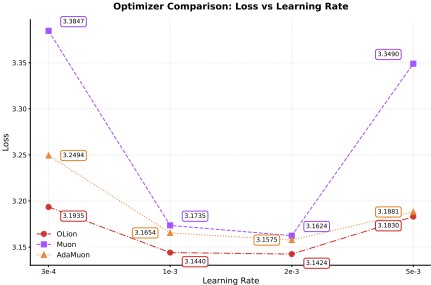

*Figure 5.* Validation loss at step 10,000 for GPT-2 small trained with different optimizers under four learning rates: $3 \times 10^{-4}$, $1 \times 10^{-3}$, $2 \times 10^{-3}$, and $5 \times 10^{-3}$. *OLion* consistently outperforms both baselines across all learning rates, demonstrating robustness to step-size selection.

### 5.3. Validating the induced implicit bias of *OLion*

To test whether *OLion* induces the implicit biases suggested by our formulation, we track both *spectral-* and $\ell_\infty$-related statistics of weight matrices during training and at convergence. We focus on GPT-2 small for a controlled study and compare AdamW, Muon, Lion, and *OLion* throughout.

**Biases along the training trajectory.** Figure 2 reports the evolution of the spectral norm and the $\ell_\infty$ norm (maximum absolute entry) for representative weight matrices of different shapes. Across all shown matrices, *OLion* maintains smaller spectral norms than AdamW and Lion, and is competitive with (often better than) Muon, consistent with the effect of the orthogonalization step on the update geometry. At the same time, *OLion* yields systematically smaller $\ell_\infty$ norms than Muon (and typically also AdamW and Lion), indicating stronger suppression of coordinate outliers—as expected from the post-orthogonalization sign operation. Together, these trends provide direct evidence that *OLion* combines spectral control and coordinate-wise control during optimization.

**Bias at the end of training.** To probe the end-of-training behavior more directly, Figure 6 summarizes the distributions of (i) singular values (SVD) and (ii) entrywise magnitudes (Abs) for the same set of matrices at convergence. Relative to AdamW, *OLion* shifts the singular-value spectrum downward, indicating a smaller effective spectral scale, while relative to Muon, *OLion* tightens the tail of absolute values, indicating smaller coordinate magnitudes. These distributional shifts match the intended combination of spectral and $\ell_\infty$ implicit biases. For additional per-layer and per-matrix case studies that further corroborate these trends, see Appendix F.

### 5.4. Extension to image pretraining: SiT

We next evaluate whether the benefits of *OLion* extend beyond language modeling to image-generation pretraining. Specifically, we pretrain SiT-B/2 (a diffusion transformer)

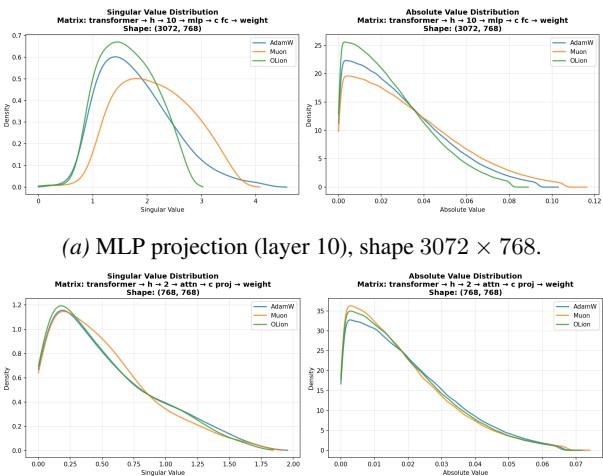

*(a)* MLP projection (layer 10), shape $3072 \times 768$.

*(b)* Attention projection (layer 2), shape $768 \times 768$.

*Figure 6.* End-of-training implicit bias on GPT-2 *Small*: distributions of singular values (SVD) and entrywise magnitudes (Abs) for representative weight matrices.

on ImageNet-1K with $256 \times 256$ resolution using the same training recipe across optimizers.

Figure 7 reports two standard objectives for this setting: the projection loss and the denoising loss. Across both metrics, *OLion* reduces loss faster than Muon throughout training. These results mirror the trends observed in language-model pretraining and suggest that the advantage of intersecting spectral and $\ell_\infty$ implicit biases is not text-specific, but carries over to diffusion-model pretraining.

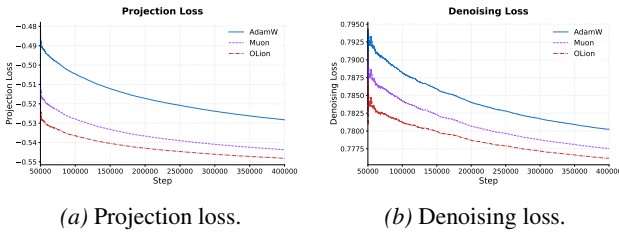

*(a)* Projection loss.  *(b)* Denoising loss.

*Figure 7.* SiT-B/2 pretraining on ImageNet-1K at $256 \times 256$. *OLion* converges faster than others on projection/denoising losses.

### 5.5. Extension to supervised fine-tuning: Llama-3.1-8B

Finally, we evaluate *OLion* on supervised fine-tuning (SFT) of Llama-3.1-8B checkpoints that were *pretrained with AdamW*. This setting is particularly relevant because prior observations suggest that Muon can suffer from a noticeable *pretrain–SFT mismatch* when applied to AdamW-pretrained models, leading to degraded downstream performance relative to AdamW (Liu et al., 2025b).

Table 1 summarizes downstream benchmark accuracies after SFT; *OLion* performs best across the listed tasks.

*Table 1.* Zero- and few-shot math benchmark accuracy (%) for the base model and SFT models fine-tuned with different optimizers. Best results are in **bold**; second-best results are in **bold gray**.

| Benchmark | Base (no SFT) | AdamW SFT | Muon SFT | *OLion* SFT |
|---|---|---|---|---|
| GSM8K 0-shot | 17.79 | **57.99** | 57.24 | **60.04** |
| GSM8K 4-shot | 52.50 | **60.35** | 58.30 | **60.58** |
| MATH 0-shot | 14.20 | 18.46 | **19.32** | **19.80** |
| NumGLUE 0-shot | 25.70 | 35.04 | **40.78** | **42.14** |
| NumGLUE 4-shot | 38.09 | 39.82 | **43.37** | **44.04** |
| SimulEq 0-shot | 12.20 | **21.48** | 21.20 | **22.57** |
| Aqua 0-shot | 23.62 | **42.52** | 41.34 | **46.45** |

One plausible explanation is that *OLion* reduces optimizer mismatch relative to Muon while still retaining spectral control. In particular, the post-orthogonalization sign step introduces an element-wise normalization structure that is closer in spirit to AdamW-style coordinate control, which may make the AdamW-pretrained checkpoint more compatible with *OLion* than with Muon's purely orthogonal (and less coordinate-normalized) update. The detailed fine-tuning setting and training curves are provided in Appendix G.

## 6. Conclusion

We presented *OLion* (*Orthogonal Lion*), a memory-efficient optimizer that combines spectral control from Muon-style orthogonalized updates with $\ell_\infty$-style coordinate control from Lion-style sign updates. Our method is motivated by a maximal-update view under norm-induced geometries: orthogonalization corresponds to a spectral geometry, while sign corresponds to an $\ell_\infty$ geometry. Composing these operations yields an efficient, single-step approximation to intersection seeking between the two geometries, with the scaled Hadamard set providing a useful idealized reference for matrix-shaped parameters.

On the theoretical side, we established that sign-after-orthogonalization preserves directional alignment with the orthogonalized direction, and provided standard convergence guarantees under non-convex smoothness conditions. Empirically, *OLion* consistently matches or outperforms AdamW and Muon across a broad range of regimes, including GPT-2 and Llama-2 language-model pretraining, SiT image pretraining, and supervised fine-tuning of Llama-3.1-8B, while reducing optimizer mismatch relative to Muon.

Several directions remain open. First, the Hadamard-intersection viewpoint suggests richer operator-splitting or multi-step intersection solvers beyond a single composed projection. Second, the sign update may offer additional benefits in distributed and low-precision settings, motivating a deeper study of communication efficiency and quantization behavior under *OLion*. We hope *OLion* provides a practical and principled step toward combining complementary implicit biases for large-scale training.

## Impact Statement

This paper proposes a new optimization method that improves training stability and efficiency for large-scale models. Our goal is to advance the optimization toolkit for modern machine learning, with potential positive impacts including reduced training compute for comparable performance and improved robustness across hyperparameter regimes, which may lower the barrier to reproducing and extending large-model training.

We do not introduce new data sources, collect personal data, or release any trained models or datasets beyond standard public resources used in the community. We will provide implementation details to support reproducibility, and we encourage responsible use consistent with existing best practices for large-model development and deployment.

## Acknowledgement

The authors would like to thank the anonymous reviewers for their valuable feedback and suggestions. The work of Zixiao Wang and Huishuai Zhang was supported in part by Beijing Major Science and Technology Project (Z251100008425004) and National Natural Science Foundation of China (Grant No. 62576015).

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

# A. Experimental Details

**Software and data availability.**    All experiments in this paper are conducted using widely adopted open-source training repositories and standard public datasets with strong community support. To facilitate reproducibility, we next provide detailed descriptions of the training setups and hyperparameter configurations used in each experiment. Our *OLion* implementation (including scripts and configurations needed to reproduce the reported results) is available at `https://github.com/kv-wang/OLion`.

## A.1. Hardware and Software Environment

All experiments are conducted with Python 3.10+ and PyTorch 2.0+ (CUDA 11.8+). For GPT-2 experiments, we use 4 NVIDIA A100 80GB GPUs with DistributedDataParallel (DDP). For Llama2-7B pretraining and Llama-3.1-8B supervised fine-tuning, we use 8 NVIDIA RTX PRO 6000 GPUs with FSDP2 (Fully Sharded Data Parallel) for distributed training. SiT-B/2 pretraining uses 4 NVIDIA RTX PRO GPUs. All experiments use mixed precision training with bfloat16 when supported; otherwise we use float32. We enable PyTorch 2.0's `torch.compile` to improve training efficiency.

## A.2. GPT-2 Pretraining Experiments

We start from the modded-nanogpt codebase by Keller Jordan (Jordan et al., 2024a) and train GPT-2 models of four sizes: Small (124M parameters), Medium (355M parameters), Large (770M parameters), and XL (1.5B parameters). The Small model has $n_{\text{layer}} = 12$, $n_{\text{head}} = 12$, and $n_{\text{embd}} = 768$. The Medium model has $n_{\text{layer}} = 24$, $n_{\text{head}} = 16$, and $n_{\text{embd}} = 1024$. The Large model has $n_{\text{layer}} = 36$, $n_{\text{head}} = 20$, and $n_{\text{embd}} = 1280$. The XL model has $n_{\text{layer}} = 48$, $n_{\text{head}} = 25$, and $n_{\text{embd}} = 1600$. We use the standard GPT-2 architecture without bias terms in LayerNorm and Linear layers, and dropout is disabled for pretraining.

We train on the OpenWebText dataset, tokenized with the standard GPT-2 tokenizer. The context length is set to 1024 tokens. We use a micro-batch size of 12 per GPU with gradient accumulation over 40 steps. We train the Small, Medium, and Large models for 20,000 steps, while the XL model is trained for 10,000 steps. The maximum learning rate is set to $6 \times 10^{-4}$ with a linear warmup for 2,000 steps, followed by a cosine decay schedule that reduces the learning rate to $6 \times 10^{-5}$. Weight decay is set to $0.1$ for all optimizers, and gradient clipping is applied at $1.0$. We evaluate every 2,000 steps using 200 evaluation iterations.

For all optimizers, we use the first-order momentum $\beta_1 = 0.9$ to align with the Adam baseline for fair comparison. AdamW uses $\beta_2 = 0.95$ and $\epsilon = 10^{-8}$. Muon uses a single momentum coefficient $\beta = 0.95$ and performs Newton-Schulz orthogonalization with $K = 5$ iterations. OLion uses $\beta_1 = 0.95$ and $\beta_2 = 0.98$ for its double-momentum scheme, with Newton-Schulz steps $K = 5$. AdaMuon uses $\beta = 0.95$ for momentum, and $K = 5$ Newton-Schulz steps.

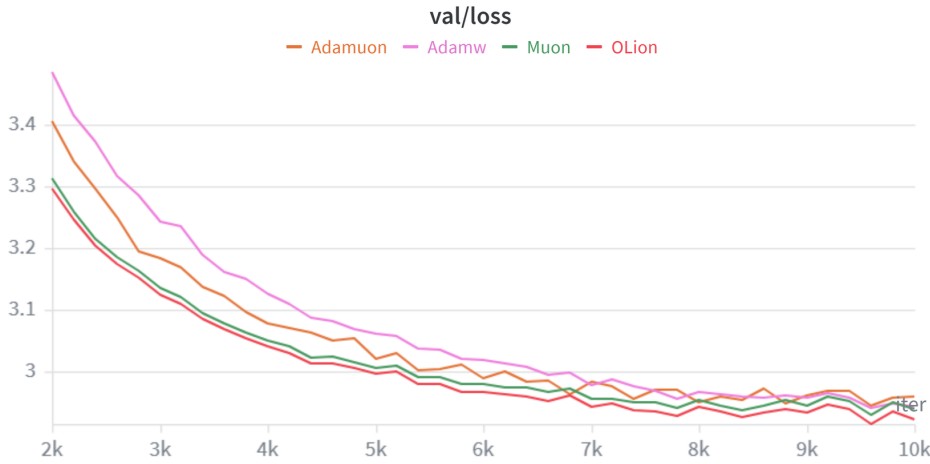

*Figure 8.* GPT-2 XL (1.5B parameters) pretraining results. *OLion* maintains the same favorable optimization trend observed for smaller GPT-2 models.

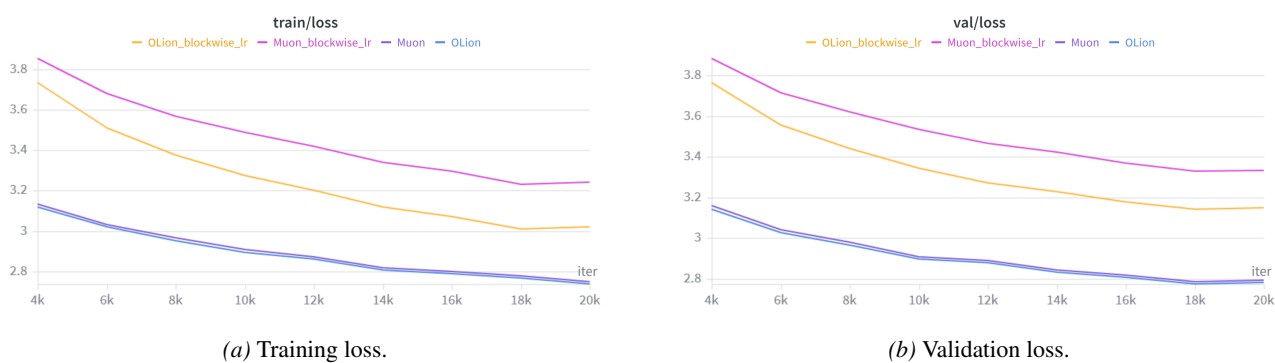

*(a)* Training loss.

*(b)* Validation loss.

*Figure 9.* Layerwise scaling ablation. The layerwise rule treats each head as a matrix and scales its learning rate by $\sqrt{d_{\text{out}}/d_{\text{in}}}$.

### A.3. Llama-7B Pretraining

We modified on the codebase of Adam-mini (Zhang et al., 2024b) and train the Llama-2-7B architecture (7 billion parameters) using the standard Llama-2 configuration with RMSNorm, SwiGLU activation, and rotary positional embeddings (RoPE). We use FSDP2 (Fully Sharded Data Parallel) for distributed training across multiple nodes. The effective global batch size is set to 4M tokens. Specifically, with a context length of 4096 tokens, a local batch size of 2, and gradient accumulation step of 64. We train for a total of 8192 steps. The maximum learning rate is $3 \times 10^{-4}$ with a linear warmup for 2,000 steps, followed by a linear decay schedule that gradually reduce the learning rate to $3 \times 10^{-5}$. Weight decay is set to $0.1$ and gradient clipping is applied at $1.0$. All training is conducted in bfloat16 mixed precision. Validation loss is computed every 100 intervals during training.

We use large-scale text datasets suitable for billion-parameter models, following the standard Llama-2 data preprocessing pipeline. Tokenization is performed using the SentencePiece tokenizer, consistent with the original Llama-2 setup. For optimizer hyperparameters, AdamW uses $\beta_1 = 0.9$, $\beta_2 = 0.95$, and $\epsilon = 10^{-8}$. Muon and AdaMuon use $\beta = 0.95$ with Newton-Schulz steps $K = 5$. OLion uses $\beta_1 = 0.95$, $\beta_2 = 0.98$, and Newton-Schulz steps $K = 5$.

### A.4. SiT-B/2 Pretraining

We train SiT-B/2 (Scalable Transformer for Diffusion Models) using the standard SiT architecture configuration. The training dataset is ImageNet-1K (Russakovsky et al., 2015) with images at $256 \times 256$ resolution, using the standard ImageNet augmentation pipeline. We use a learning rate of $1 \times 10^{-4}$ and train for 400,000 steps with a cosine decay learning rate schedule. The batch size is adjusted per GPU memory constraints. Weight decay is set to $0.1$ and gradient clipping is applied at $1.0$. Training is conducted in bfloat16 or float16 mixed precision. We track both projection loss and denoising loss separately, using the standard diffusion projection loss and denoising loss formulations. For optimizer hyperparameters, Muon uses $\beta = 0.95$ with Newton-Schulz steps $K = 5$, and OLion uses $\beta_1 = 0.95$, $\beta_2 = 0.98$, and Newton-Schulz steps $K = 5$.

## B. Diagonal-isotropy Verification for a Random Gaussian Model

In this section we introduce a standard random model for the singular vectors $(\mathbf{U}, \mathbf{V})$, and show that the diagonal-isotropy condition in Assumption 4.2 holds with high probability with an explicit parameter $\varepsilon$.

### B.1. Random model

Let $r \geq 2$ and $d_1, d_2 \geq r$. We generate $(\mathbf{U}, \mathbf{V})$ as follows:

1. Sample $\mathbf{A} \in \mathbb{R}^{d_1 \times r}$ and $\mathbf{B} \in \mathbb{R}^{d_2 \times r}$ with i.i.d. $\mathcal{N}(0, 1)$ entries, independently.

2. Let $\mathbf{U} = \text{qf}(\mathbf{A}) \in \mathbb{R}^{d_1 \times r}$ and $\mathbf{V} = \text{qf}(\mathbf{B}) \in \mathbb{R}^{d_2 \times r}$ be the orthonormal $Q$-factors in the QR decompositions of $\mathbf{A}$ and $\mathbf{B}$.

Then $\mathbf{U}$ and $\mathbf{V}$ are independent Haar-distributed random matrices on the Stiefel manifolds (i.e., their column spaces are

uniformly random), and satisfy $\mathbf{U}^\top \mathbf{U} = \mathbf{I}_r$, $\mathbf{V}^\top \mathbf{V} = \mathbf{I}_r$.

Define

$$\mathbf{Q} := \mathbf{U}\mathbf{V}^\top \in \mathbb{R}^{d_1 \times d_2}, \qquad \mathbf{Z} := \mathrm{sign}(\mathbf{Q}) \ \ (\text{entrywise, with } \mathrm{sign}(0) = 0).$$

We also define the diagonal correlation vector

$$\mathbf{m} := \mathrm{diag}(\mathbf{U}^\top \mathbf{Z}\mathbf{V}) \in \mathbb{R}^r, \qquad \bar{m} := \frac{1}{r}\mathbf{1}^\top \mathbf{m}.$$

A key identity is

$$\mathbf{1}^\top \mathbf{m} = \mathrm{tr}(\mathbf{U}^\top \mathbf{Z}\mathbf{V}) = \langle \mathbf{U}\mathbf{V}^\top, \mathbf{Z} \rangle = \langle \mathbf{Q}, \mathrm{sign}(\mathbf{Q}) \rangle = \|\mathbf{Q}\|_1, \qquad \Rightarrow \qquad \bar{m} = \frac{\|\mathbf{Q}\|_1}{r}. \tag{14}$$

## B.2. Main claim: diagonal-isotropy holds with high probability

**Theorem B.1** (Diagonal-isotropy for random Gaussian singular vectors). *Let $(\mathbf{U}, \mathbf{V})$ follow the random model in Section B.1, and define $\mathbf{Q} = \mathbf{U}\mathbf{V}^\top$ and $\mathbf{Z} = \mathrm{sign}(\mathbf{Q})$. Then there exist absolute constants $c, C > 0$ such that, with probability at least*

$$1 - 2r^{-10} - 2\exp(-c\,d_1 d_2),$$

*the following two events hold simultaneously:*

$$\left\|\mathrm{diag}(\mathbf{U}^\top \mathbf{Z}\mathbf{V}) - \frac{\|\mathbf{Q}\|_1}{r}\mathbf{1}\right\|_2 \leq C\sqrt{r \log r}, \tag{15}$$

$$\|\mathbf{Q}\|_1 \geq c\sqrt{r\,d_1 d_2}. \tag{16}$$

*Consequently, on the same event, Assumption 4.2 holds with*

$$\varepsilon = \frac{C}{c}\sqrt{\frac{r \log r}{d_1 d_2}}. \tag{17}$$

*In particular, if $d_1 d_2 \gg r \log r$, then $\varepsilon = o(1)$.*

## B.3. Proof of Theorem B.1

**Step 1: Centering factor.** By Equation (14), we have the exact relation

$$\bar{m} = \frac{1}{r}\mathbf{1}^\top \mathrm{diag}(\mathbf{U}^\top \mathbf{Z}\mathbf{V}) = \frac{\|\mathbf{Q}\|_1}{r},$$

which matches the centering used in Assumption 4.2. Moreover, by Haar symmetry, the coordinates of $\mathbf{m}$ are exchangeable (permuting columns of $\mathbf{U}$ and $\mathbf{V}$ does not change the joint law), so it is natural that each coordinate concentrates at the same scale.

**Step 2: A lower bound on $\|\mathbf{Q}\|_1$.** Under independent Haar $\mathbf{U}, \mathbf{V}$, each entry

$$Q_{ij} = \sum_{k=1}^r U_{ik}V_{jk}$$

is mean-zero and, in the high-dimensional regime, behaves approximately as a Gaussian with variance $\mathrm{Var}(Q_{ij}) \approx r/(d_1 d_2)$ (a standard consequence of rotational invariance and the fact that each $U_{ik}, V_{jk}$ is of size $\Theta(d_1^{-1/2}), \Theta(d_2^{-1/2})$). Thus $\mathbb{E}|Q_{ij}| \approx \sqrt{\frac{2}{\pi}}\sqrt{\frac{r}{d_1 d_2}}$ and

$$\mathbb{E}\|\mathbf{Q}\|_1 = \sum_{i,j} \mathbb{E}|Q_{ij}| \approx \sqrt{\frac{2}{\pi}}\sqrt{r\,d_1 d_2}.$$

Furthermore, $\|\mathbf{Q}\|_1$ concentrates around its mean (as a Lipschitz function of the underlying Gaussian matrices $(\mathbf{A}, \mathbf{B})$ away from negligible ill-conditioning events), yielding the high-probability bound Equation (16) for an absolute constant $c > 0$.

**Step 3: Concentration of diagonal correlations.** For each $k \in [r]$,

$$m_k = (\mathbf{U}^\top \mathbf{Z} \mathbf{V})_{kk} = u_k^\top \mathbf{Z} v_k = \sum_{i=1}^{d_1} \sum_{j=1}^{d_2} U_{ik} V_{jk} \, \text{sign}(Q_{ij}).$$

Heuristically, $(Q_{ij}, U_{ik}V_{jk})$ is close to a jointly Gaussian pair with

$$\text{Var}(Q_{ij}) \approx \frac{r}{d_1 d_2}, \qquad \text{Var}(U_{ik}V_{jk}) \approx \frac{1}{d_1 d_2}, \qquad \text{Cov}(Q_{ij}, U_{ik}V_{jk}) \approx \frac{1}{d_1 d_2}.$$

In particular, each summand has variance of order $1/(d_1 d_2)$ and the sum over $d_1 d_2$ terms yields constant-scale fluctuations for $m_k$. Formally, one may replace $\text{sign}(\cdot)$ by a smooth approximation (e.g. $\tanh(\cdot/\tau)$), apply Gaussian concentration to obtain sub-Gaussian tails for the smoothed $m_k$, and then remove the smoothing using anti-concentration of $Q_{ij}$ near zero. This yields an absolute constant $c_0 > 0$ such that

$$\mathbb{P}(|m_k - \mathbb{E}m_k| \geq t) \leq 2\exp(-c_0 t^2), \qquad \forall t \geq 0.$$

Taking a union bound over $k = 1, \ldots, r$ gives, with probability at least $1 - 2r^{-10}$,

$$\max_{k \in [r]} |m_k - \mathbb{E}m_k| \; \leq \; C_0 \sqrt{\log r}$$

for an absolute constant $C_0 > 0$, hence

$$\|\mathbf{m} - \mathbb{E}\mathbf{m}\|_2 \leq \sqrt{r} \max_{k \in [r]} |m_k - \mathbb{E}m_k| \leq C_0 \sqrt{r \log r}.$$

Finally, since $\bar{m} = \|\mathbf{Q}\|_1 / r$ is the empirical mean of $(m_k)$ by Equation (14) and $\|\mathbf{Q}\|_1$ concentrates (Step 2), the deviation between centering by $\mathbb{E}m_k$ and by $\bar{m}$ is of the same (or smaller) order. Absorbing constants yields Equation (15) with an absolute constant $C$.

**Step 4: Conclude Assumption 4.2.** On the event where Equation (15) and Equation (16) both hold,

$$\left\| \text{diag}(\mathbf{U}^\top \text{sign}(\mathbf{U}\mathbf{V}^\top)\mathbf{V}) - \frac{\|\mathbf{U}\mathbf{V}^\top\|_1}{r} \mathbf{1} \right\|_2 \leq C\sqrt{r \log r} = \frac{C}{c} \sqrt{\frac{r \log r}{d_1 d_2}} \cdot \frac{\|\mathbf{Q}\|_1}{\sqrt{r}},$$

which is exactly Assumption 4.2 with $\varepsilon$ as in Equation (17). This completes the proof.

## C. Diagonal-Isotropy Verification Along the Training Trajectory

Empirically, we verify that the matrix updates along the training trajectory satisfy Assumption 4.2 with small values of $\varepsilon$. Concretely, during GPT-2 small (124M) pretraining we record the value of $\varepsilon$ (i.e., the smallest constant such that the diagonal-isotropy inequality holds) for the gradient-related matrix $\widetilde{\mathbf{G}}_t$ at each update, under three optimizers: AdamW, Muon, and OLion. The results are shown in Figure 10a and Figure 10b.

## D. Proofs of Lemma 4.3

*Proof.* Let $\mathbf{Q} := \mathbf{U}\mathbf{V}^\top$ and $\mathbf{S} := \text{sign}(\mathbf{Q})$ (entrywise, with $\text{sign}(0) = 0$). Since $\mathbf{Z} = \mathbf{U}\mathbf{\Sigma}\mathbf{V}^\top$ and $\alpha = \text{tr}(\mathbf{\Sigma})/r$, we have

$$\mathbf{Z} - \alpha \mathbf{U}\mathbf{V}^\top = \mathbf{U}(\mathbf{\Sigma} - \alpha \mathbf{I})\mathbf{V}^\top.$$

Using the Frobenius inner product $\langle \mathbf{A}, \mathbf{B} \rangle = \text{tr}(\mathbf{A}^\top \mathbf{B})$ and the orthonormality $\mathbf{U}^\top \mathbf{U} = \mathbf{I}_r$, $\mathbf{V}^\top \mathbf{V} = \mathbf{I}_r$, we obtain

$$\langle \mathbf{Z} - \alpha \mathbf{U}\mathbf{V}^\top, \; \mathbf{S} \rangle = \langle \mathbf{U}(\mathbf{\Sigma} - \alpha \mathbf{I})\mathbf{V}^\top, \; \mathbf{S} \rangle = \text{tr}\Big((\mathbf{\Sigma} - \alpha \mathbf{I})\mathbf{U}^\top \mathbf{S} \mathbf{V}\Big).$$

Since $\mathbf{\Sigma} - \alpha \mathbf{I}$ is diagonal, only the diagonal entries of $\mathbf{U}^\top \mathbf{S}\mathbf{V}$ contribute to the trace. Define

$$\mathbf{m} := \text{diag}(\mathbf{U}^\top \mathbf{S}\mathbf{V}) \in \mathbb{R}^r, \qquad \beta := \frac{1}{r}\mathbf{1}^\top \mathbf{m}.$$

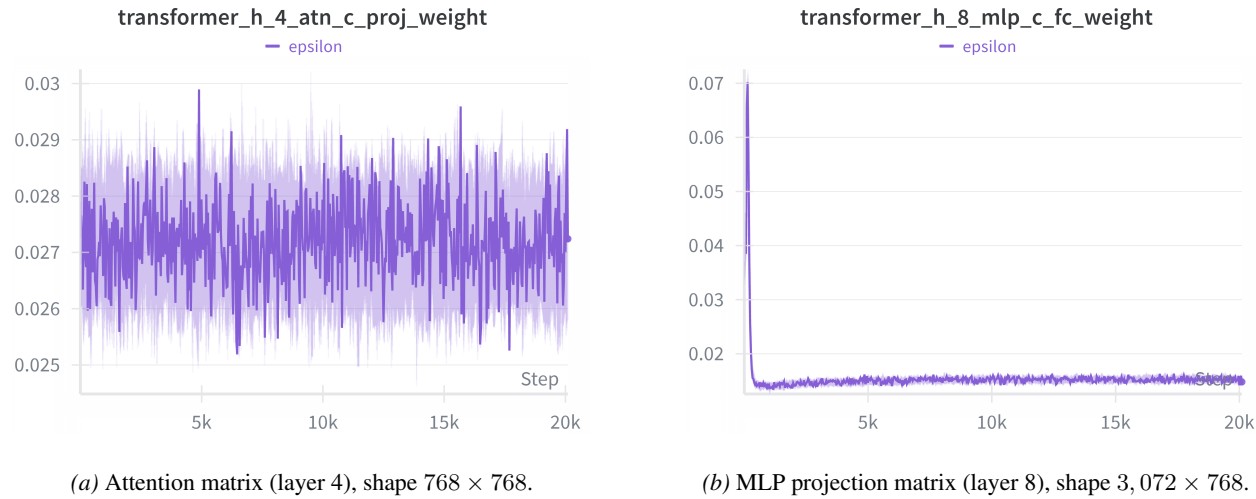

*(a)* Attention matrix (layer 4), shape $768 \times 768$.

*(b)* MLP projection matrix (layer 8), shape $3,072 \times 768$.

*Figure 10.* The $\varepsilon$ values are consistently small for different matrices across the training procedure.

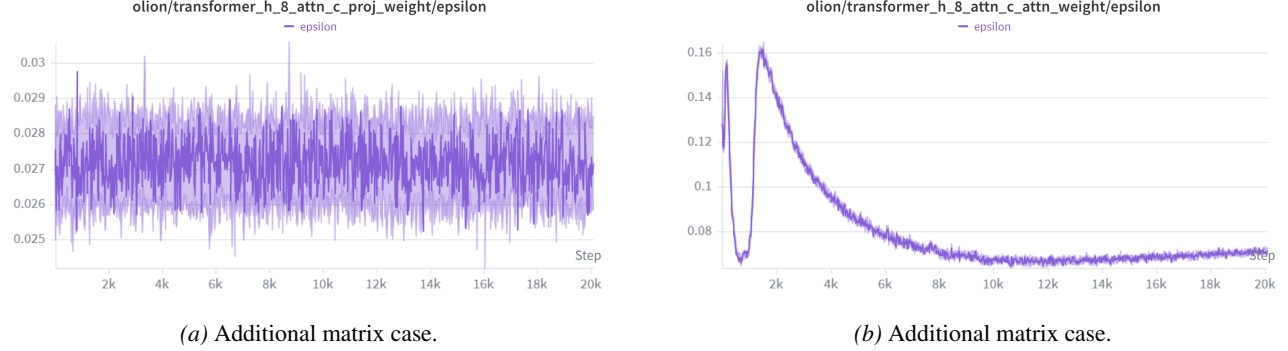

*(a)* Additional matrix case.

*(b)* Additional matrix case.

*Figure 11.* Additional diagonal-isotropy verification along the training trajectory. As in Figure 10, the $\varepsilon$ values remain small across matrix choices and optimizers.

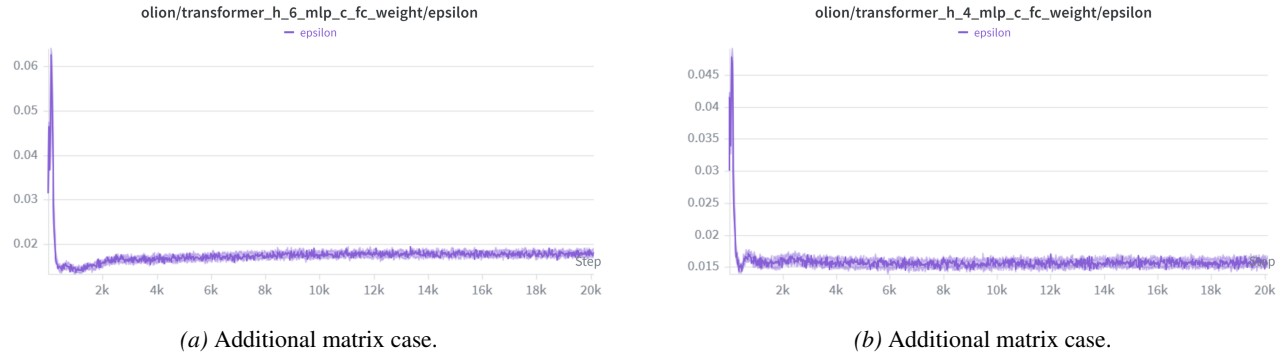

*(a)* Additional matrix case.

*(b)* Additional matrix case.

*Figure 12.* Additional diagonal-isotropy verification along the training trajectory. The measured $\varepsilon$ values provide further evidence that Assumption 4.2 is empirically reasonable during training.

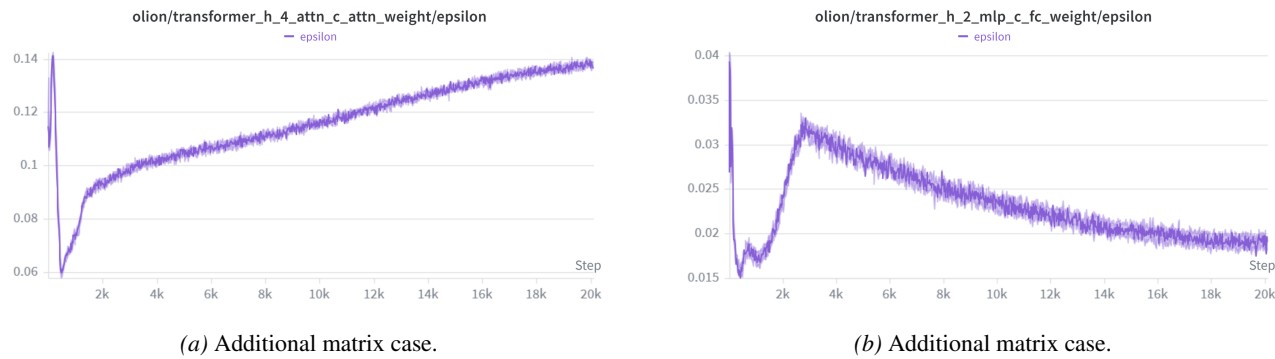

*(a)* Additional matrix case.         *(b)* Additional matrix case.

*Figure 13.* Additional diagonal-isotropy verification along the training trajectory. The same small-$\varepsilon$ pattern persists across more sampled matrices.

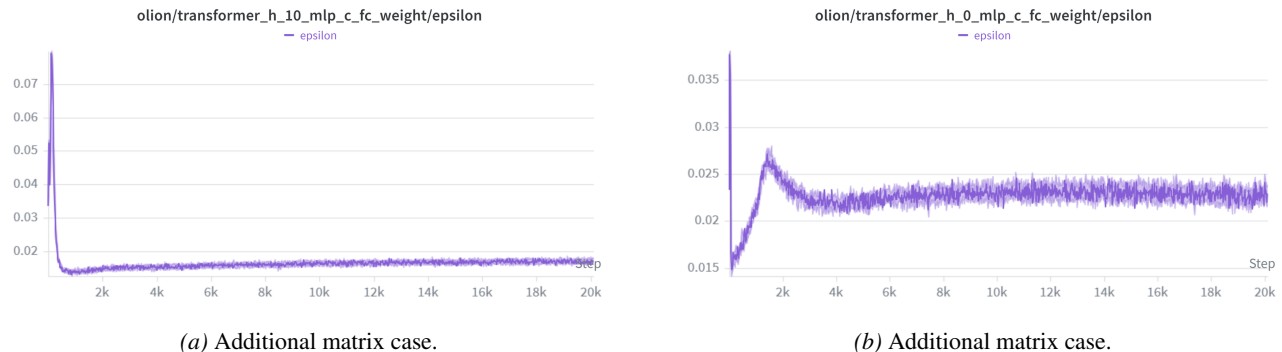

*(a)* Additional matrix case.         *(b)* Additional matrix case.

*Figure 14.* Additional diagonal-isotropy verification along the training trajectory. These cases further support the stability of the diagonal-isotropy behavior.

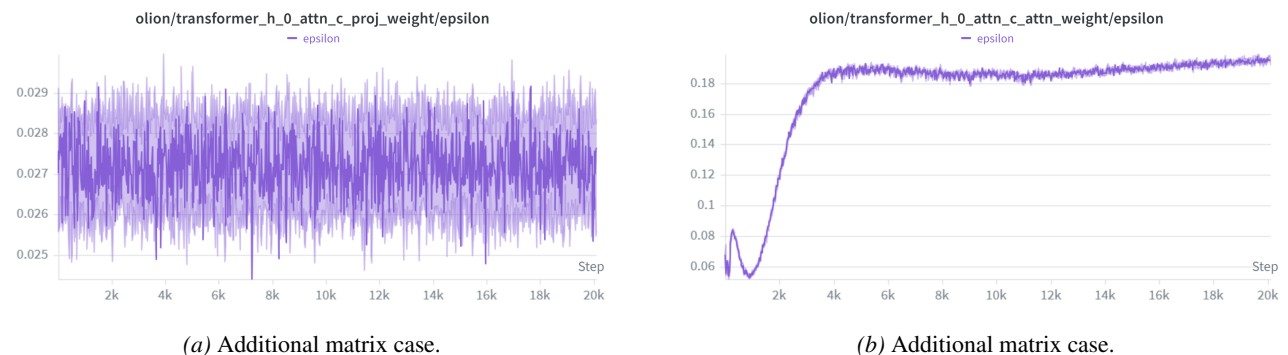

*(a)* Additional matrix case.         *(b)* Additional matrix case.

*Figure 15.* Additional diagonal-isotropy verification along the training trajectory. Together with Figure 10, these results show that the $\varepsilon$ values are consistently small across a broader collection of matrices.

Then

$$\mathrm{tr}\Big((\boldsymbol{\Sigma} - \alpha\boldsymbol{I})\,\boldsymbol{U}^\top\boldsymbol{S}\,\boldsymbol{V}\Big) = \sum_{k=1}^{r}(\sigma_k - \alpha)\,m_k = \langle\boldsymbol{\sigma} - \alpha\mathbf{1},\ \mathbf{m}\rangle,$$

where $\boldsymbol{\sigma} = (\sigma_1,\ldots,\sigma_r)^\top$ and $m_k$ is the $k$-th entry of $\mathbf{m}$.

We next relate $\beta$ to $\|\boldsymbol{Q}\|_1$. By cyclicity of trace,

$$\mathbf{1}^\top\mathbf{m} = \mathrm{tr}(\boldsymbol{U}^\top\boldsymbol{S}\boldsymbol{V}) = \mathrm{tr}(\boldsymbol{V}\boldsymbol{U}^\top\boldsymbol{S}) = \langle\boldsymbol{U}\boldsymbol{V}^\top,\ \boldsymbol{S}\rangle = \langle\boldsymbol{Q},\ \mathrm{sign}(\boldsymbol{Q})\rangle = \|\boldsymbol{Q}\|_1,$$

hence $\beta = \|\boldsymbol{U}\boldsymbol{V}^\top\|_1/r$.

Now decompose $\mathbf{m} = \beta\mathbf{1} + (\mathbf{m} - \beta\mathbf{1})$:

$$\langle\boldsymbol{\sigma} - \alpha\mathbf{1},\ \mathbf{m}\rangle = \beta\,\langle\boldsymbol{\sigma} - \alpha\mathbf{1},\ \mathbf{1}\rangle + \langle\boldsymbol{\sigma} - \alpha\mathbf{1},\ \mathbf{m} - \beta\mathbf{1}\rangle.$$

Because $\alpha = \mathrm{tr}(\boldsymbol{\Sigma})/r$, we have $\langle\boldsymbol{\sigma} - \alpha\mathbf{1},\ \mathbf{1}\rangle = \mathrm{tr}(\boldsymbol{\Sigma}) - \alpha r = 0$, so the first term vanishes. Therefore,

$$\left|\langle\boldsymbol{Z} - \alpha\boldsymbol{U}\boldsymbol{V}^\top,\ \mathrm{sign}(\boldsymbol{U}\boldsymbol{V}^\top)\rangle\right|$$
$$= \left|\langle\boldsymbol{\sigma} - \alpha\mathbf{1},\ \mathbf{m} - \beta\mathbf{1}\rangle\right|$$
$$\le \|\boldsymbol{\sigma} - \alpha\mathbf{1}\|_2\,\|\mathbf{m} - \beta\mathbf{1}\|_2.$$

Noting that $\|\boldsymbol{\sigma} - \alpha\mathbf{1}\|_2 = \|\boldsymbol{\Sigma} - \alpha\boldsymbol{I}\|_F$, and applying Assumption 4.2 (i.e., $\|\mathbf{m} - \beta\mathbf{1}\|_2 \le \varepsilon\,\|\boldsymbol{U}\boldsymbol{V}^\top\|_1/\sqrt{r}$), we conclude

$$\left|\langle\boldsymbol{Z} - \alpha\boldsymbol{U}\boldsymbol{V}^\top,\ \mathrm{sign}(\boldsymbol{U}\boldsymbol{V}^\top)\rangle\right| \le \varepsilon\,\frac{\|\boldsymbol{U}\boldsymbol{V}^\top\|_1}{\sqrt{r}}\,\Big\|\boldsymbol{\Sigma} - \alpha\boldsymbol{I}\Big\|_F,$$

as desired. $\qquad\square$

### D.1. Discussion on Lemma 4.3

**Why the cancellation-aware estimate (Lemma 4.3) matters.** A naïve bound to control $\langle\mathbf{Z} - \alpha\mathbf{Q},\mathbf{S}\rangle$ is given by Cauchy-Schwarz:

$$\left|\langle\mathbf{Z} - \alpha\mathbf{Q},\mathbf{S}\rangle\right| \le \|\mathbf{Z} - \alpha\mathbf{Q}\|_F\,\|\mathbf{S}\|_F,$$

but this bound ignores two key structural facts: (i) $\alpha = \mathrm{tr}(\boldsymbol{\Sigma})/r$ makes $\boldsymbol{\Sigma} - \alpha\mathbf{I}$ *trace* 0, so its positive and negative diagonal deviations must cancel, and (ii) $\mathbf{S} = \mathrm{sign}(\mathbf{Q})$ is not an arbitrary matrix, but is tightly coupled to $\mathbf{Q} = \mathcal{O}(\mathbf{Z})$. We exploit this structure by rewriting

$$\langle\mathbf{Z} - \alpha\mathbf{Q},\mathbf{S}\rangle = \mathrm{tr}\Big((\boldsymbol{\Sigma} - \alpha\mathbf{I})\,\mathbf{U}^\top\mathbf{S}\mathbf{V}\Big) = \sum_{j=1}^{r}(\sigma_j - \alpha)\,m_j$$

where $\mathbf{m} := \mathrm{diag}(\mathbf{U}^\top\mathbf{S}\mathbf{V})$ and then centering $\mathbf{m}$ by its mean $\bar{m} = \|\mathbf{Q}\|_1/r$, which is admissible because $\sum_j(\sigma_j - \alpha) = 0$. This yields

$$\langle\mathbf{Z} - \alpha\mathbf{Q},\mathbf{S}\rangle = \sum_{j=1}^{r}(\sigma_j - \alpha)\,(m_j - \bar{m}),$$

so the magnitude is governed by *fluctuations* of the diagonal correlations $m_j$ around their average, rather than by the ambient size $\|\mathbf{S}\|_F = \Theta(\sqrt{d_1 d_2})$. Under the diagonal-isotropy assumption, these fluctuations are small, giving the sharper bound

$$\left|\langle\mathbf{Z} - \alpha\mathbf{Q},\mathbf{S}\rangle\right| \lesssim \|\boldsymbol{\Sigma} - \alpha\mathbf{I}\|_F \cdot \varepsilon\frac{\|\mathbf{Q}\|_1}{\sqrt{r}},$$

which can be orders of magnitude tighter than Cauchy–Schwarz bound when $\mathbf{Q}$ is dense (large $\|\mathbf{Q}\|_1$) and the diagonal correlations are nearly uniform.

## E. Proof of Theorem 4.4

*Proof.* By $L$-smoothness (Assumption 4.1),

$$f(\boldsymbol{X}_{t+1}) \leq f(\boldsymbol{X}_t) + \langle \boldsymbol{G}_t, \boldsymbol{X}_{t+1} - \boldsymbol{X}_t \rangle + \frac{L}{2} \|\boldsymbol{X}_{t+1} - \boldsymbol{X}_t\|_F^2.$$

Using $\boldsymbol{X}_{t+1} - \boldsymbol{X}_t = -\eta_t \boldsymbol{S}_t$ gives

$$f(\boldsymbol{X}_{t+1}) \leq f(\boldsymbol{X}_t) - \eta_t \langle \boldsymbol{G}_t, \boldsymbol{S}_t \rangle + \frac{L}{2} \eta_t^2 \|\boldsymbol{S}_t\|_F^2. \tag{18}$$

Next, since $\boldsymbol{S}_t = \text{sign}(\boldsymbol{Q}_t)$ with $\boldsymbol{Q}_t \in \mathbb{R}^{d_1 \times d_2}$,

$$\|\boldsymbol{S}_t\|_F^2 = \|\text{sign}(\boldsymbol{Q}_t)\|_F^2 = d_1 d_2.$$

It remains to lower bound $\langle \boldsymbol{G}_t, \boldsymbol{S}_t \rangle$. Let $\boldsymbol{G}_t = \boldsymbol{U}_t \boldsymbol{\Sigma}_t \boldsymbol{V}_t^\top$ have rank $r_t$, and let $\boldsymbol{Q}_t = \mathcal{O}(\boldsymbol{G}_t) = \boldsymbol{U}_t \boldsymbol{V}_t^\top$. Then

$$\langle \boldsymbol{G}_t, \boldsymbol{S}_t \rangle = \langle \boldsymbol{G}_t, \text{sign}(\boldsymbol{Q}_t) \rangle$$
$$= \Big( \langle \alpha_t \boldsymbol{Q}_t, \text{sign}(\boldsymbol{Q}_t) \rangle + \langle \boldsymbol{G}_t - \alpha_t \boldsymbol{Q}_t, \text{sign}(\boldsymbol{Q}_t) \rangle \Big).$$

The first term equals $\alpha_t \|\boldsymbol{Q}_t\|_1$ because $\langle \boldsymbol{Q}_t, \text{sign}(\boldsymbol{Q}_t) \rangle = \|\boldsymbol{Q}_t\|_1$. For the second term, Lemma 4.3 yields

$$\left| \langle \boldsymbol{G}_t - \alpha_t \boldsymbol{Q}_t, \text{sign}(\boldsymbol{Q}_t) \rangle \right| \leq \varepsilon \frac{\|\boldsymbol{Q}_t\|_1}{\sqrt{r_t}} \left\| \boldsymbol{\Sigma}_t - \alpha_t \boldsymbol{I} \right\|_F = \varepsilon \|\boldsymbol{Q}_t\|_1 \alpha_t \rho_t.$$

Therefore,

$$\langle \boldsymbol{G}_t, \boldsymbol{S}_t \rangle \geq \|\boldsymbol{Q}_t\|_1 \alpha_t (1 - \varepsilon \rho_t) = \Phi_t.$$

Plugging this and $\|\boldsymbol{S}_t\|_F^2 = d_1 d_2$ into Equation (18) gives Equation (12). Summing Equation (12) from $t = 0$ to $T - 1$ and using $f(\boldsymbol{X}_T) \geq f_{\inf}$ yields Equation (13). $\qquad\square$

## F. Implicit Bias Verification

We sample several individual matrices from different layers and examine their singular-value and absolute-value distributions in detail. Figures 16, 17, 18, and 19 show representative MLP projection matrices (layers 7, 8, and 10) and an attention projection matrix (layer 2), respectively. Consistent with the aggregate statistics, *OLion* produces more concentrated singular-value distributions (with lower leading components) than AdamW, while also producing tighter absolute-value distributions (with smaller element magnitudes) than Muon. This confirms that the combined spectral and $\ell_\infty$ biases manifest robustly across individual layers and matrix shapes.

## G. Llama-3.1-8B Supervised Fine-Tuning Details

We perform full parameter fine-tuning on Llama-3.1-8B model that was pretrained with AdamW, using the codebase of torchtitan (Liang et al., 2025). We do not use LoRA or other parameter-efficient fine-tuning methods. The training dataset is MathInstruct, formatted in the standard instruction-following format for mathematical reasoning tasks. We use a learning rate of $3 \times 10^{-5}$, which is typical for supervised fine-tuning, with a linear warmup for 100 steps followed by cosine decay that gradually reduces learning rate to $3 \times 10^{-6}$. The trainer is initialized with a local batch size of 8 per GPU, global batch size of 512, and gradient accumulation over 16 steps. The sequence length is set to 2048 tokens. We train for a total of 1533 steps. Weight decay is set to 0.1 and gradient clipping is applied at 1.0. All training is conducted in bfloat16 mixed precision.

We evaluate the fine-tuned models on five mathematical reasoning benchmarks: GSM8K (grade school math word problems) in both 0-shot and 4-shot settings, MATH (competition-level math problems) in 0-shot setting, NumGLUE (numerical reasoning benchmark) in both 0-shot and 4-shot settings, SimulEq (simultaneous equations) in 0-shot setting, and Aqua (arithmetic reasoning) in 0-shot setting. We use standard evaluation scripts for each benchmark. For optimizer

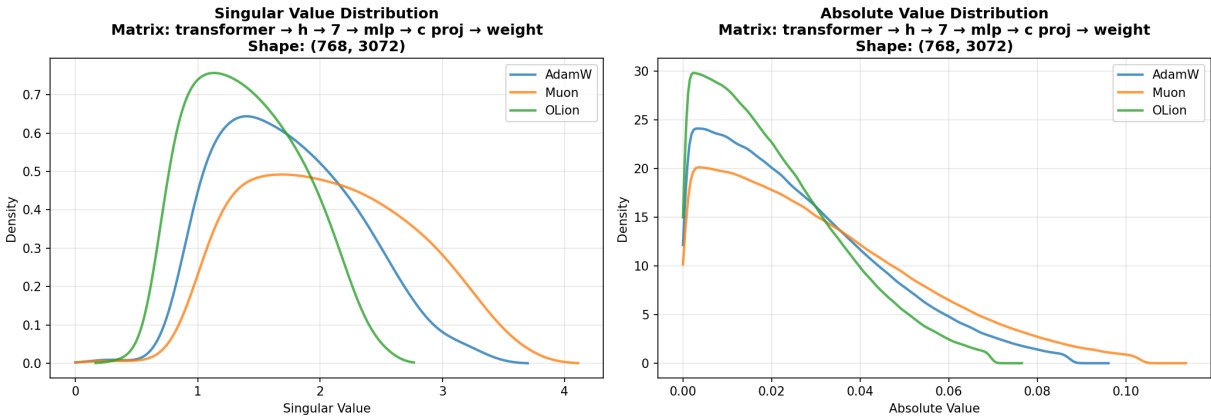

*Figure 16.* Singular value and absolute value distributions for the MLP projection weight matrix at layer 7 (shape $768 \times 3072$).

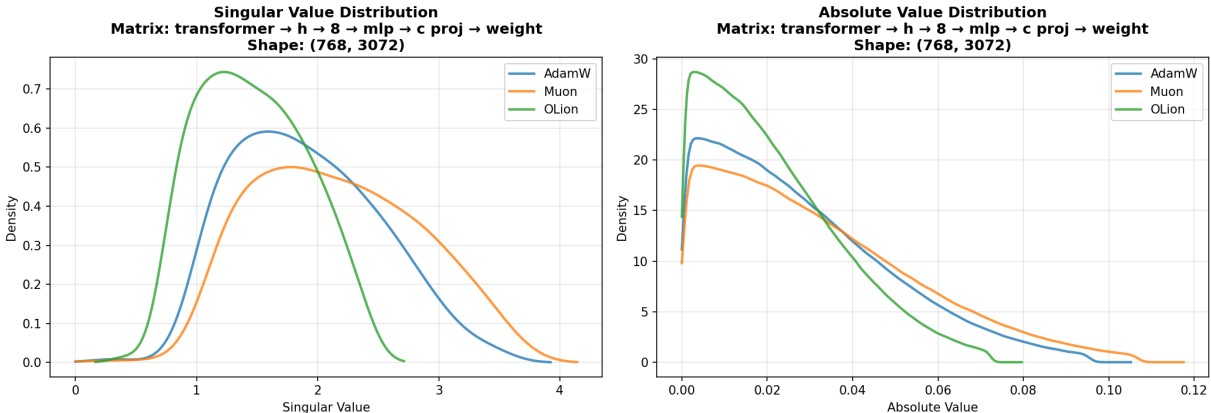

*Figure 17.* Singular value and absolute value distributions for the MLP projection weight matrix at layer 8 (shape $768 \times 3072$).

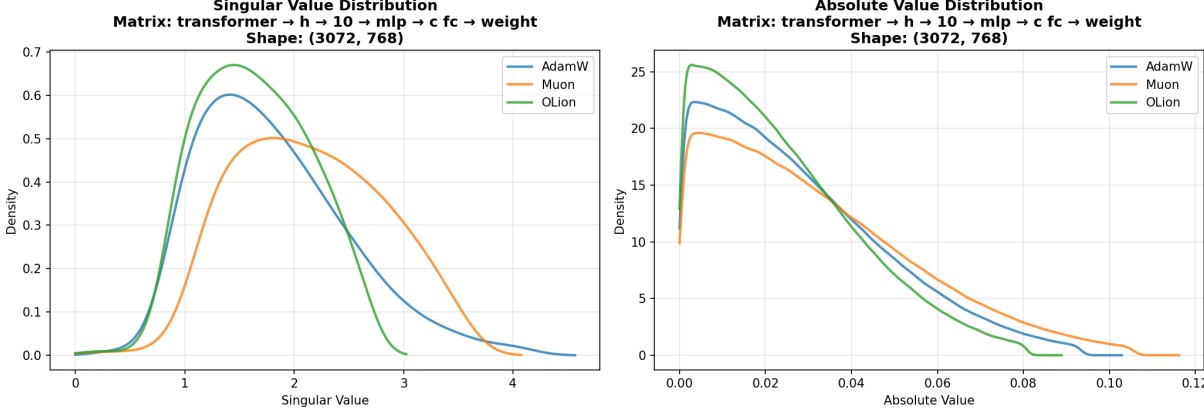

*Figure 18.* Singular value and absolute value distributions for the MLP projection weight matrix at layer 10 (shape $3072 \times 768$).

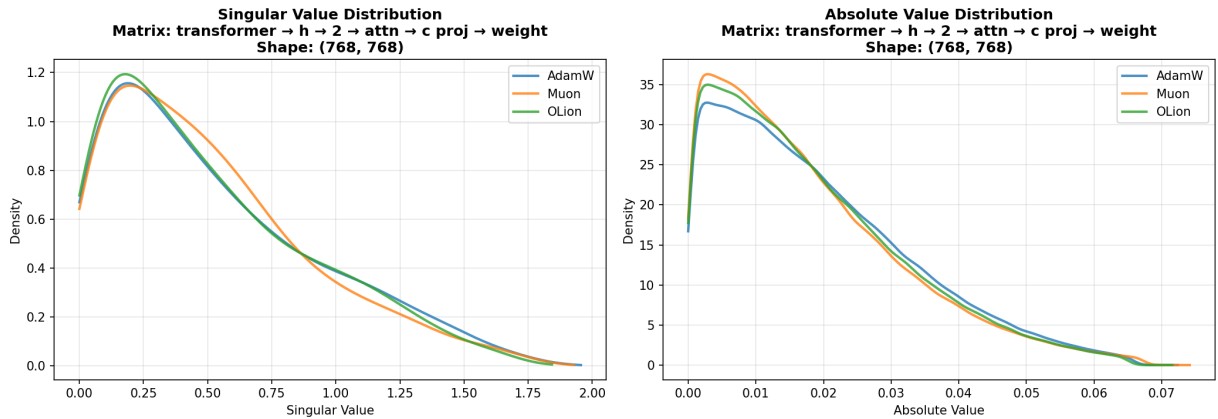

*Figure 19.* Singular value and absolute value distributions for the attention projection weight matrix at layer 2 (shape $768 \times 768$).

hyperparameters, AdamW uses $\beta_1 = 0.9$, $\beta_2 = 0.95$, and $\epsilon = 10^{-8}$. Muon uses $\beta = 0.95$ with Newton-Schulz steps $K = 5$. OLion uses $\beta_1 = 0.95$, $\beta_2 = 0.98$, and Newton-Schulz steps $K = 5$.

Figure 20 reports SFT loss trajectories. *OLion* consistently improves over Muon and is competitive with (and in some runs better than) AdamW throughout training.

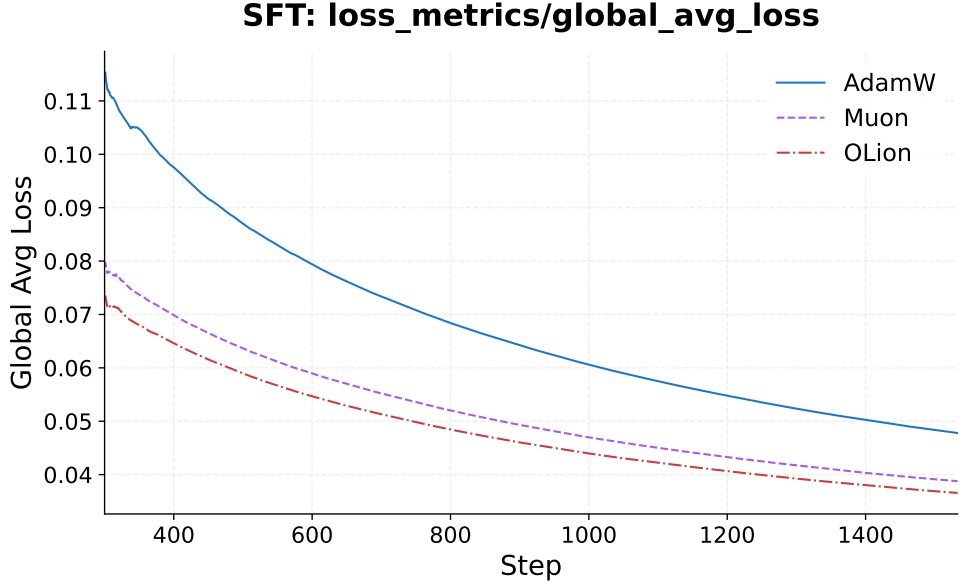

*Figure 20.* Supervised fine-tuning loss on Llama-3.1-8B.

