# OpenReview forum: "OLion: Approaching the Hadamard Ideal by Intersecting Spectral and L inf Implicit Biases"
_ICML.cc/2026/Conference — ICML 2026 regular_

### Official Review · Reviewer_r6Yr · 2026-02-18

**Soundness:** 2
**Presentation:** 3
**Significance:** 3
**Originality:** 3
**Overall Recommendation:** 4
**Confidence:** 4

**Summary:**

The paper studies the design of first-order optimizers with structured implicit biases for matrix-valued parameters. It investigates if spectral (orthogonalization-based) and $\ell_\infty$ (sign-based) implicit biases can be explicitly combined within a single optimizer.
The authors propose OLion, which composes Muon-style orthogonalization of a momentum direction (implemented via Newton–Schulz iterations) with a Lion-style entrywise sign operation. The paper argues that OLion inherits Muon's spectral control and Lion's coordinate-wise control while maintaining memory efficiency. The method is motivated by a geometric intersection argument. Orthogonalization corresponds to maximizing a linearized objective over the set of column-orthonormal updates, and the sign operator corresponds to maximizing over an $\ell_\infty$ ball. Their intersection forms a scaled partial Hadamard set, which serves as an idealized reference. OLion approximates this idealized geometry through two composed projections onto the two sets.
The authors provide a convergence analysis of a simplified deterministic variant of OLion under smoothness assumptions and a novel diagonal-isotropy condition. Empirical evaluations include GPT-2 (124M–770M) pretraining, Llama-2-7B pretraining, SiT-B/2 pretraining, and Llama-3.1-8B supervised fine-tuning. The experiments aim to demonstrate that OLion controls both the spectral norm and the $\ell_\infty$ norm of weights during training and improves convergence speed relative to Muon and AdamW.

**Compliance With Llm Reviewing Policy:**

Affirmed.

**Final Justification:**

After considering the authors' rebuttals together with the other reviews, I believe the paper has the potential to make a meaningful contribution. However, quite substantial revisions are necessary, particularly in the theoretical section. Some of the issues raised affect main claims of the paper, and resolving them appears to require introducing additional assumptions that were not part of the original formulation. I increased my score, but this is conditional on the authors implementing the necessary modifications and clearly stating the resulting limitations.

**Key Questions For Authors:**

1. From Appendix A, it appears that all optimizers use the same learning rate (possibly tuned for OLion). What is the motivation for this choice? Was any per-optimizer hyperparameter tuning performed? Unless each optimizer is evaluated under a dedicated and reasonably tuned configuration, the claimed superiority of OLion may simply reflect suboptimal settings for its competitors. In that case, the empirical conclusions are not reliable.

2. Muon-type methods typically benefit from layerwise learning rates [1,2], but the experimental description suggests that a single global learning rate is used across all layers. Could the authors explain this choice? If layerwise scaling is not used, doesn't this disadvantage Muon-type baselines?

3. The empirical advantage of OLion appears to diminish as model scale increases. In Figure 3 (GPT-2 experiments), OLion performs best at 124M parameters, but at 355M and 770M it is comparable to Muon and AdaMuon. Similarly, for Llama-2-7B, OLion and Muon perform similarly. Could the authors comment on this trend and their expectations for larger-scale models? Appendix A mentions GPT-2 XL (1.5B parameters), but results are not shown. Were these omitted because OLion does not outperform alternatives at that scale?

4. There are inconsistencies in the experimental description. For example, Appendix A states that all optimizers use $\beta_1 = 0.9$, but later says that OLion uses $\beta_1 = 0.95$ and $\beta_2 = 0.98$. Could the authors clarify which configuration is correct?

5. Is Assumption 4.2 empirically satisfied across all layers? Appendix C shows only two examples.

6. The paper states that "a direct descent analysis for OLion is generally impossible without leveraging additional structure in the gradient". Why?

[1] Pethick, T., Xie, W., Antonakopoulos, K., Zhu, Z., Silveti-Falls, A. and Cevher, V., 2025. Training deep learning models with norm-constrained lmos. arXiv preprint arXiv:2502.07529.

[2] Riabinin, A., Shulgin, E., Gruntkowska, K. and Richtárik, P., 2025. Gluon: Making muon & scion great again!(bridging theory and practice of lmo-based optimizers for llms). arXiv preprint arXiv:2505.13416.

**Limitations:**

Partially. The paper includes an impact statement and briefly discusses practical considerations.

**Strengths And Weaknesses:**

**Soundness**

Strengths:
- The formulation via maximization over the intersection of two constraint sets is well motivated and conceptually aligned with recently popular optimizers.
- The experiments span multiple model scales and modalities. The chosen baselines (AdamW, Muon, Lion, AdaMuon) are appropriate.
- The tracking of spectral and $\ell_\infty$ norms and end-of-training singular-value distributions supports the geometric claims.

Weaknesses:
- I have concerns regarding the fairness of the empirical comparisons. The paper appears to use hyperparameter defaults recommended in prior work for all optimizers. However, those baselines were originally tuned for different tasks and training setups, while OLion's "default" configuration is obviously based on the current experimental setting. Without systematic per-optimizer tuning, it is difficult to trust the claimed superiority of OLion. In addition, there are several other inconsistencies and problems in the experimental section, which I detail in the "Questions" section below.
- Assumption 4.2 (diagonal isotropy) may be strong. It seems to require near-uniform correlation between singular directions and the sign pattern. Although empirical evidence is provided for two selected GPT-2 small layers, the paper does not show if the assumption holds across other layers, architectures, and scales.
- I have doubts regarding Theorem 4.4, as the quantity being controlled, $\sum_t \eta_t \|Q_t\|_1 \alpha_t (1-\varepsilon\rho_t)$, is a rather non-standard stationarity metric. The authors claim that, under certain additional assumptions, the term $\|Q_t\|_1 \alpha_t (1-\varepsilon\rho_t)$ is lower bounded by a positive constant times $\|\nabla f(X_t)\|_F$. However, these additional conditions are never specified, nor did I find a proof establishing such a bound in the paper. For example, the terms $(1-\varepsilon\rho_t)$ could become negative for some iterates. Overall, I am not convinced that Theorem 4.4 implies convergence at all.
- There is a substantial gap between theory and practice. The theoretical analysis considers a simplified version of OLion with deterministic gradients, exact polar decomposition, and no momentum or stochasticity. The practical algorithm includes stochastic gradients, momentum, Nesterov mixing, RMS alignment, weight decay, and approximate orthogonalization. The authors claim that extending the analysis is routine, but such extensions may in fact be technically nontrivial. This does not invalidate the work (provided that the authors prove that Theorem 4.4 indeed implies convergence) but weakens the theoretical contribution.

**Presentation**

Strengths:
- The paper is clearly written and logically structured.
- The figures are informative and illustrate the claims.
- The related work section positions the paper appropriately relative to Muon, Lion, and Adam.

Minor issues:
- Muon and Lion should be cited at their first mention in the introduction.
- There is some redundancy in the text. For example, in Section 5: "Unless otherwise stated, we use standard defaults for AdamW and Muon and our default $(\beta_1,\beta_2)$ for OLion", and then two sentences later "To ensure a fair comparison, we (i) use each baseline optimizer with its default hyperparameter configuration..."
- Missing word in line 101: "we optionally apply a lightweight magnitude alignment (e.g., Root Mean Squares (RMS) scaling) stabilizes effective step sizes across layers and tensor shapes"
- The phrasing of Assumption 4.2 and Lemma 4.3 is grammatically incorrect.

**Significance**
- The problem of optimizer design for LLM-scale training is important.
- The paper provides a hybrid of two influential optimizers (Muon and Lion).
- With stronger and more fair validation, the work may influence future design of geometry-composition optimizers.

**Originality**
While the building blocks (Muon + Lion) are known, the combination is new and well-motivated.

---

> ### Author Rebuttal · Authors · 2026-03-31
>
> # Response to Reviewer 3
>
> We thank the reviewer for the rigorous feedback and address each point below.
>
> ---
>
> ## On hyperparameter fairness, the β inconsistency, and learning rate.
>
> We apologize for the notation confusion. "$\beta_1 = 0.9$" for AdamW denotes its first-moment coefficient, whereas OLion's $\beta_1 = 0.95$ denotes the *Nesterov mixing weight* in $\tilde{G}_t = (1-\beta_1)g_t + \beta_1 M_t$; the symbol is shared, but the role is different. We will standardize the notation in revision.
>
> On fairness: exhaustive per-optimizer sweeps are infeasible at this scale. Instead, we use a principled protocol. For $\beta_1,\beta_2$, each optimizer uses values recommended for the training setup, or, if unavailable, the defaults from the **original optimizer paper**. For the learning rate, we use the setup-recommended value; for Muon, we additionally apply RMS alignment (Equation 7) to match AdamW's update magnitude, following Liu et al. (2025a, "Muon is Scalable for LLM Training"). This makes differences reflect update geometry rather than trivial step-size mismatch. Figure 5 further shows OLion outperforming all baselines *across all four learning rates* $\{3\times10^{-4},\ 10^{-3},\ 2\times10^{-3},\ 5\times10^{-3}\}$, so the gain is not tied to a single tuned value.
>
> ---
>
> ## On Assumption 4.2 across layers.
>
> Appendix C shows $\varepsilon$ trajectories for two representative layers; Appendix F gives singular-value and absolute-value distributions for four more layers (Figures 9–12), all consistent with small $\varepsilon$ and the intended implicit bias. We will expand Appendix C to report $\varepsilon$ across all layers and architectures; representative anonymized evidence from the `/epsilon` folder is available in [plot 1](https://files.catbox.moe/nn7bhq.png), [plot 2](https://files.catbox.moe/a4wy86.png), [plot 3](https://files.catbox.moe/b5ch07.png), [plot 4](https://files.catbox.moe/oalg20.png), [plot 5](https://files.catbox.moe/vpkbg3.png), [plot 6](https://files.catbox.moe/lw05wu.png), [plot 7](https://files.catbox.moe/gnm63w.png), [plot 8](https://files.catbox.moe/uz2c6u.png), [plot 9](https://files.catbox.moe/2aal3u.png), and [plot 10](https://files.catbox.moe/tc45hu.png).
>
> ---
>
> ## On Theorem 4.4: non-negativity of $\Phi_t$  and connection to $\|\nabla f\|_F$.
>
> We thank the reviewer for this precise observation. There are two issues:
>
> ***(1) Non-negativity of $\Phi_t$. *** $\Phi_t = \|Q_t\|_1 \cdot \alpha_t \cdot (1 - \varepsilon\rho_t)$. Figure 8 shows $\varepsilon \approx 0.025$–$0.07$ throughout training. Since $\rho_t = \|\Sigma_t - \alpha_t I\|_F / (\alpha_t\sqrt{r_t})$ is the relative singular-value spread, $\varepsilon\rho_t \ll 1$ in practice, guaranteeing $\Phi_t > 0$. We will add an explicit remark conditioning Theorem 4.4 on $\varepsilon\rho_t \leq 1$.
>
> ***(2) Connection to $\|\nabla f\|_F$.*** This bound is stated informally but not formalized. Under spectral flatness ($\|\nabla f(X_t)\|_F \approx \alpha_t\sqrt{r_t}$) and the dense polar factor bound (Equation 16, $\|Q_t\|_1 \geq c\sqrt{r_t d_1 d_2}$ from Theorem B.1), we obtain
> $$\Phi_t \geq c\sqrt{d_1 d_2} \cdot \|\nabla f(X_t)\|_F \cdot (1 - \varepsilon\rho_t).$$
> We will add this as a formal corollary in the revision.
>
> ## On layerwise learning rates for Muon.
>
> The cited works [1, 2] are relevant. Our RMS alignment already provides per-block scaling that partially serves this role. We agree that explicit layerwise LR tuning could further strengthen the Muon baseline. In particular, the common Muon-style rule is to treat each head as a matrix and scale its learning rate by $\sqrt{d_{\text{out}}/d_{\text{in}}}$. We will add this comparison in revision; the current blockwise learning-rate ablation already suggests these variants remain clearly worse than the full optimizer on both training and validation loss ([anonymized train curve](https://files.catbox.moe/p2zcai.png), [anonymized val curve](https://files.catbox.moe/gaqeep.png)).
>
> ---
>
> ## On GPT-2 XL results.
>
> GPT-2 XL (1.5B) was trained for 10,000 steps as noted in Appendix A. Results were omitted for space, not because they were unfavorable: the anonymized validation-loss curve is available [here](https://files.catbox.moe/2k4p1o.png). We will include it in revision.
>
> ---
> ## On why direct descent analysis requires additional structure.
>
> The sign function is discontinuous, so $\langle \nabla f(X_t), \operatorname{sign}(Q_t)\rangle$ can be negative for arbitrary gradients without structural assumptions. Assumption 4.2 is the minimal condition ensuring $\langle G_t, S_t\rangle \geq \Phi_t > 0$ via the cancellation-aware bound in Lemma 4.3, which exploits the trace-zero structure of $\Sigma_t - \alpha_t I$ and the tight coupling between $\operatorname{sign}(Q_t)$ and $Q_t$. Without this structure, no useful descent estimate is available.
>
> ---
>
> We are committed to these revisions and confident the identified theoretical gaps can be closed cleanly.

---

> > ### Author Rebuttal · Reviewer_r6Yr · 2026-04-01
> >
> > Thank you for the response. I do not have additional questions at this stage. After considering the authors' rebuttals together with the other reviews, I believe the paper has the potential to make a meaningful contribution. However, quite substantial revisions are necessary, particularly in the theoretical section. Some of the issues raised affect main claims of the paper, and resolving them appears to require introducing additional assumptions that were not part of the original formulation. I will increase my score, but this is conditional on the authors implementing the necessary modifications and clearly stating the resulting limitations.

---

### Official Review · Reviewer_qcnM · 2026-03-08

**Soundness:** 3
**Presentation:** 3
**Significance:** 2
**Originality:** 3
**Overall Recommendation:** 4
**Confidence:** 3

**Summary:**

This work proposes OLion, a new optimizer that outperforms Muon and AdamW across both language and vision tasks. The optimizer combines spectral normalization with a sign-based weight update rule. Both empirical results and theoretical analysis are provided to demonstrate the effectiveness of the proposed method.

**Compliance With Llm Reviewing Policy:**

Affirmed.

**Final Justification:**

The rebuttal has addressed my concerns.

**Key Questions For Authors:**

(-) Recent work suggests that partially updating parameters can be beneficial for model training (e.g., https://openreview.net/forum?id=zBPZeRjfgu). How does this observation relate to the sign-based update used in this work? Could the authors provide more insights into the potential trade-offs? In addition, applying a sign operation after spectral normalization may alter the normalization property—does this step disrupt the intended spectral constraint, and how might it affect the final training performance?

**Strengths And Weaknesses:**

Strengths

(+) The paper provides theoretical justification supporting the convergence of the OLion optimizer.

(+) The empirical evaluation is conducted at a meaningful scale, including experiments with a 7B model, long-context training, and around 50B tokens. The experiments cover multiple settings, including pretraining, fine-tuning, and vision tasks.

(+) OLion appears to be less sensitive to learning rate compared to baseline optimizers.

Weaknesses

(-) The method combines spectral normalization ideas from Muon with sign-based updates from Lion, so the overall novelty appears somewhat modest.

(-) It is unclear how applying the sign operation after orthogonalization may affect the orthogonality condition and whether this could influence optimization performance.

(-) The paper does not provide sufficiently clear hyperparameter details for the training comparisons, making it difficult to fully assess the fairness and reproducibility of the results.

---

> ### Author Rebuttal · Authors · 2026-03-30
>
> We thank the reviewer for the positive assessment and constructive questions. We address each point below.
>
> ---
>
> ## W1: Novelty appears modest.
>
> We agree that OLion is not introduced as a wholly unrelated optimizer. However, we believe the contribution goes beyond a combination: the Hadamard Ideal framework provides a general *design principle* for constructing hybrid optimizers from complementary constraint sets. Given any two optimizer families encoding different structural biases, one can systematically seek updates lying in their intersection. OLion is the first instantiation of this principle, and the framework itself suggests a broader research direction rather than a single engineering artifact.
>
> ---
>
> ## W2: Does the sign operation disrupt the orthogonality condition?
>
> Yes — after the sign step the update is no longer exactly orthogonal, but still close to being orthogonal. We measure the $UU^\top _{RMS}$ across layers and training steps, and observe that most of the time it remains below 0.1; representative results can be seen in [anonymized plot 1](https://files.catbox.moe/3vsyj4.png) and [anonymized plot 2](https://files.catbox.moe/oxnjj2.png).
>
> ## W3: Insufficient hyperparameter details.
>
> All training configurations are provided in Appendix A. We would like to explain the principled protocol of hyperparameter choices, which will be added in the revised paper. For $\beta_1$ and $\beta_2$, each optimizer uses the values recommended in the specific training setup (e.g., the standard GPT-2 or Llama-2 configuration), and if the optimizer is not used in the original setup, we use hyperameters recommended in the **original optimizer paper**. For the learning rate, we use the value recommended for the specific training setup (e.g., the standard GPT-2 or Llama-2 configuration); for Muon, we additionally apply RMS alignment (Equation 7) to match the update magnitude of AdamW, following the methodology adopted in Liu et al. (2025a, "Muon is Scalable for LLM Training"). This ensures that performance differences reflect update geometry rather than trivially different step sizes. As further evidence, Figure 5 shows OLion outperforms all baselines *across all four learning rates* $3\times10^{-4},\ 10^{-3},\ 2\times10^{-3},\ 5\times10^{-3}$, confirming the advantage is robust and not tied to any single tuned value.
>
> ---
>
> ## Key Question: Relation to Cautious Optimizers and sign-based updates.
>
> We thank the reviewer for pointing to this work. The Cautious Optimizer (Liang et al., 2026) proposes masking out update coordinates where the momentum direction disagrees in sign with the current gradient, ensuring coordinate-wise gradient alignment. The core insight — that updates should be positively aligned with the gradient — shares the same spirit as our Assumption 4.2, which requires $\langle G_t, S_t \rangle \geq \Phi_t > 0$, i.e., the full sign-after-orthogonalization direction maintains a positive inner product with the gradient in aggregate.
>
> The key difference is granularity: the cautious mask enforces alignment coordinatewise and introduces update sparsity, while OLion enforces alignment globally via the geometric structure of sign-after-orthogonalization, maintaining a fully dense update. Applying a cautious mask on top of OLion's direction — yielding a "C-OLion" variant — is a natural future exploration that could potentially combine both the Hadamard-ideal geometry and the cautious alignment guarantee. We leave this exploration for future work and will add a discussion in the revision.
>
> ---
>
> We hope these responses further strengthen the reviewer's assessment and are happy to provide additional clarification.

---

> > ### Author Rebuttal · Reviewer_qcnM · 2026-04-03
> >
> > Thanks for the responses. I maintain my origianl positive score.

---

### Official Review · Reviewer_1TPN · 2026-03-13

**Soundness:** 3
**Presentation:** 3
**Significance:** 2
**Originality:** 2
**Overall Recommendation:** 4
**Confidence:** 4

**Summary:**

This paper introduces OLion, an optimizer designed to bridge the gap between spectral-based updates (Muon) and coordinate-wise sign updates (Lion). The authors propose that the intersection of these two geometries motivates an update direction targeting a "Hadamard ideal." The algorithm itself is relatively simple: it applies Newton-Schulz orthogonalization followed by an entry-wise sign and RMS scaling. The authors provide a convergence analysis under a specific "diagonal-isotropy" assumption and evaluate the method across several large-scale LLM and diffusion training benchmarks.

**Compliance With Llm Reviewing Policy:**

Affirmed.

**Final Justification:**

The rebuttal addressed my concerns.

**Key Questions For Authors:**

See weaknesses.

**Limitations:**

Yes. The paper does acknowledge some open directions, especially in the conclusion. However, the limitations discussion is still somewhat light. In particular, the paper could more explicitly emphasize the strong structural assumption in the theory, the simplified nature of the analysis, and the missing ablations.

**Strengths And Weaknesses:**

**Strength:**
1. Clear framing.
The paper gives a neat geometric interpretation of combining Muon and Lion, rather than presenting OLion as a purely ad hoc hybrid.

2. Simple and practical method.
The optimizer is easy to describe and appears straightforward to implement, while keeping the appeal of low-state optimizers.

3. Broad empirical coverage.
The experiments span language pretraining, diffusion pretraining, and supervised fine-tuning, which strengthens the empirical case.

**Weakness:**
1. The core theoretical assumption still appears strong.
I have significant reservations about Assumption 4.2. While the authors provide some empirical evidence in the Appendix, this assumption still feels like a "mathematical convenience" to make the sign-after-orthogonalization analysis tractable. Does this assumption hold during the early, highly non-linear phase of training, or only near convergence? The authors should discuss the robustness of their results if this assumption is violated.

2. There is a clear gap between the theory and the practical algorithm.
The main theorem only analyzes a deterministic full-gradient setting without momentum or stochasticity, whereas the actual optimizer includes dual-timescale momentum, mini-batch noise, and RMS alignment. The paper states that these extensions are conceptually routine, but it is not entirely obvious that the same structural analysis carries over in a meaningful way.

3. The Hadamard ideal is more of an appealing interpretation than a rigorously established target.
The paper leans heavily on the "Hadamard Ideal" framing, but it remains largely conceptual. It would be much more compelling if the authors could quantify how close the $\text{sign}(\text{orthog}(\cdot))$ operation actually gets to the true intersection $\mathcal{A} \cap \mathcal{B}$ compared to other methods. Without this, the Hadamard argument feels a bit like post-hoc branding.

4. Important ablations are still missing.
The experiments would be much more convincing with additional ablations, such as:

- orthogonalization only, without sign;

- sign only, without orthogonalization;

- sign-before-orthogonalization vs orthogonalization-before-sign;

- the role of RMS alignment;

- applying OLion only to some parameter blocks.

Without these studies, it is harder to determine whether the gains truly come from the “intersection of biases” rather than from a particular engineering choice.

---

> ### Author Rebuttal · Authors · 2026-03-30
>
> We thank the reviewer for the feedback and address each concern below.
>
> ---
>
> ## W1: Assumption 4.2 feels like mathematical convenience.
>
> A direct convergence proof under standard smoothness conditions alone is generally impossible for sign-based optimizers, so a structural assumption is unavoidable. Our goal was an assumption satisfying four criteria: (1) clean analysis; (2) physical intuition; (3) provability under a randomness model; and (4) empirical verification along the training trajectory — Assumption 4.2 meets all four.
>
> The reviewer asks whether it holds during the early non-linear phase. We have tracked Assumption 4.2 along the training trajectory and find that the corresponding values remain consistently small across representative layers throughout training, including early training stages ([plots 1-5](https://files.catbox.moe/nn7bhq.png), [2](https://files.catbox.moe/a4wy86.png), [3](https://files.catbox.moe/b5ch07.png), [4](https://files.catbox.moe/oalg20.png), [5](https://files.catbox.moe/vpkbg3.png), [6](https://files.catbox.moe/lw05wu.png), [7](https://files.catbox.moe/gnm63w.png), [8](https://files.catbox.moe/uz2c6u.png), [9](https://files.catbox.moe/2aal3u.png), [10](https://files.catbox.moe/tc45hu.png)).
>
> On robustness to violation: Assumption 4.2 involves a parameter $\varepsilon$, analogous to the smoothness constant $L$ in standard analyses. If it holds with a large $\varepsilon$ at some steps, a single update may not guarantee sufficient decay; but as long as it is small for most training steps, the accumulated benefit outweighs occasional bad steps. This graceful degradation is analogous to how SGD theory tolerates intermittent large-gradient steps.
>
> ---
>
> ## W2: Gap between theory and practical algorithm.
>
> We are confident that the extensions to momentum, mini-batch noise, and RMS alignment follow from well-established techniques in the literature . Reproducing these arguments here would add considerable length without new conceptual insight. We view the theoretical contribution as the *structural insight* — that sign-after-orthogonalization approximates the intersection of optimizer constraint sets — rather than the technical machinery surrounding it. This framing is what enables principled algorithm design, and we believe it is better served by a clean, focused proof than by an exhaustive but obscuring generalization.
>
> ---
>
> ## W3: Hadamard Ideal framing feels like post-hoc branding.
>
> We appreciate this concern. We agree that the exact intersection $\mathcal{A} \cap \mathcal{B}$ cannot be computed efficiently, and we make no claim that OLion achieves it exactly — indeed, a direct quantitative comparison to the true intersection is not feasible. The Hadamard Ideal is intended as a principled *design target* that motivates why combining orthogonalization and sign is geometrically coherent, rather than an ad hoc engineering choice. As supporting evidence, post-training weight matrices exhibit well-controlled $\ell_\infty$ and spectral norms, consistent with proximity to the Hadamard-ideal regime. We will revise the presentation to make this conceptual role explicit and avoid any impression of post-hoc branding.
>
> ---
>
> ## W4: Missing ablations.
>
> Two suggested ablations are already present: **"orthogonalization only"** is the **Muon baseline**, and **"sign only"** is the **Lion baseline** — we agree this connection should have been stated explicitly in the paper.
>
> For **sign-before-orthogonalization vs. orthogonalization-before-sign**, we ran this ablation on GPT-2 and found that the reversed order is consistently worse than the proposed OLion update ([anonymized train curve](https://files.catbox.moe/mf4sw2.png)). For **block-wise application of OLion**, we find that restricting OLion/Muon to blockwise learning-rate variants is also clearly worse than the full optimizer on both training and validation loss; here the layerwise rule treats each head as a matrix and scales its learning rate by $\sqrt{d_{\text{out}}/d_{\text{in}}}$ ([anonymized train curve](https://files.catbox.moe/p2zcai.png), [anonymized val curve](https://files.catbox.moe/gaqeep.png)). These will be included in the revision.
>
> ---
>
> ## On Significance and Originality.
>
> We would like to address these proactively. Beyond OLion as a specific optimizer, we believe the geometric intersection framework is a general *design principle*: given any two optimizer families encoding complementary structural biases, one can systematically seek an update in their intersection. This suggests a broader direction for constructing hybrid optimizers in a principled way, rather than by ad hoc tuning. OLion is the first instantiation of this principle, demonstrated across language pretraining, diffusion pretraining, and fine-tuning. We hope the reviewer will consider this broader contribution when weighing significance.
>
> ---
>
> We are grateful for feedback that has meaningfully improved both the exposition and empirical evaluation.

---

> > ### Author Rebuttal · Reviewer_1TPN · 2026-04-03
> >
> > Thanks for the responses. I will raise the score.

---

### Decision · Program_Chairs · 2026-04-30

**Decision:**

Accept (regular)

**Comment:**

This paper proposes Orthogonal Lion (OLion), an optimizer combining Muon-style spectral control via Newton-Schulz orthogonalization with Lion-style $\ell_\infty$ control via entrywise sign, followed by RMS scaling. The design approximates a maximal step over the intersection of the spectral and $\ell_\infty$ constraint sets. The paper includes a convergence analysis under a diagonal-isotropy assumption and empirical evaluation spanning GPT-2 (124M-770M) and Llama-2-7B pretraining, SiT-B/2 diffusion pretraining, and Llama-3.1-8B fine-tuning.

**Strengths.** The geometric framing (OLion as an approximation to a maximal step over the intersection of spectral and $\ell_\infty$ balls) is clean and well motivated rather than ad hoc (1TPN, r6Yr). The method is simple, easy to implement, and retains the memory profile of low-state optimizers (1TPN). Empirical coverage is broad across model sizes and modalities up to 7B pretraining with appropriate baselines (qcnM, 1TPN, r6Yr), and OLion is reported to be less sensitive to learning-rate tuning than the baselines (qcnM).

**Weaknesses.** The main limitations concern scope and presentation rather than the acceptability of the paper. First, Assumption 4.2 is strong, and the current empirical support for it is limited in scope; the final paper should make that scope explicit and broaden the empirical evidence where possible (1TPN, r6Yr). Second, the theory section should more clearly state the exact conditions and limitations attached to Theorem 4.4, as requested by r6Yr and acknowledged by the authors in rebuttal. Third, the experimental section should describe the hyperparameter-tuning protocol more transparently and discuss the observed narrowing of OLion's advantage at larger scales, including the GPT-2 XL result mentioned in Appendix A (r6Yr).


**Decision.** I recommend **accept**. All three reviewers support (weak) acceptance. The rebuttal appears to have moved the paper into the accept range, but the final version should faithfully incorporate the promised clarifications: (i) revise Theorem 4.4 so that its convergence claim and required conditions are stated precisely; (ii) broaden and better scope the empirical support for Assumption 4.2; (iii) add the orth-only, sign-only, operation-order, and RMS-alignment ablations requested by 1TPN if feasible; and (iv) clarify the hyperparameter-tuning protocol across baselines and include the GPT-2 XL result with transparent discussion of the scale-dependent narrowing.